# *SlCCD1A* Enhances the Aroma Quality of Tomato Fruits by Promoting the Synthesis of Carotenoid-Derived Volatiles

**DOI:** 10.3390/foods10112678

**Published:** 2021-11-03

**Authors:** Guo-Ting Cheng, Yu-Shun Li, Shi-Ming Qi, Jin Wang, Pan Zhao, Qian-Qi Lou, Yan-Feng Wang, Xiang-Qian Zhang, Yan Liang

**Affiliations:** 1College of Horticulture, Northwest A&F University, Xianyang 712100, China; chengguoting@nwafu.edu.cn (G.-T.C.); liyushun2016@nwafu.edu.cn (Y.-S.L.); qishiming2008@nwafu.edu.cn (S.-M.Q.); jw6127@nwafu.edu.cn (J.W.); zhaopan2019@nwafu.edu.cn (P.Z.); qqlou1996@nwafu.edu.cn (Q.-Q.L.); 2State Key Laboratory of Crop Stress Biology in Arid Regions, Northwest A&F University, Xianyang 712100, China; 3Shaanxi Key Laboratory of Chinese Jujube, Yan’an University, Yan’an 716000, China; wyf@yau.edu.cn (Y.-F.W.); xiangqian_zhang@163.com (X.-Q.Z.); 4College of Life Science, Yan’an University, Yan’an 716000, China

**Keywords:** *Solanum lycopersicum*, flavor, volatiles, carotenoid, *SlCCD1A*

## Abstract

The loss of volatiles results in the deterioration of flavor in tomatoes. Volatiles are mainly derived from fatty acid, carotenoid, phenylpropane, and branched chain amino acids. In this study, the tomato accession CI1005 with a strong odor and accession TI4001 with a weak odor were analyzed. The volatile contents were measured in tomato fruits using gas chromatography-mass spectrometry. The scores of tomato taste and odor characteristics were evaluated according to hedonistic taste and olfaction. It was found that the content of fatty acid-derived volatiles accounted for more than half of the total volatiles that had grassy and fatty aromas. Phenylpropane-derived volatiles had irritation and floral aromas. Branched-chain amino acid-derived volatiles had a caramel aroma. Carotenoid-derived volatiles had floral, fruity, fatty, and sweet-like aromas, preferred by consumers. A lack of carotenoid-derived volatiles affected the flavor quality of tomato fruits. The accumulation of carotenoid-derived volatiles is regulated by carotenoid cleavage oxygenases (CCDs). A tissue-specific expression analysis of the *SlCCD* genes revealed that the expression levels of *SlCCD1A* and *SlCCD1B* were higher in tomato fruits than in other tissues. The expression levels of *SlCCD1A* and *SlCCD1B* were consistent with the trend of the carotenoid-derived volatile contents. The expression of *SlCCD1A* was higher than that for *SlCCD1B*. A bioinformatics analysis revealed that SlCCD1A was more closely linked to carotenoid metabolism than SlCCD1B. The overexpression of *SlCCD1A* indicated that it could cleave lycopene, α-carotene, and β-carotene to produce 6-methyl-5-hepten-2-one, geranylacetone, α-ionone, and β-ionone, increasing the floral, fruity, fatty, and sweet-like aromas of tomato fruits. The flavor quality of tomato fruits could be improved by overexpressing *SlCCD1A*.

## 1. Introduction

Tomatoes (*Solanum lycopersicum*) are an important dual-purpose product considered to be both a vegetable and fruit. In 2019, the global tomato production reached 180.77 million tons [1]. The flavor and quality of tomatoes has not improved with the increase in yield but, instead, decreased in modern cultivated tomatoes [2,3,4]. This is due to the significantly reduced contents of glucose, fructose, citric acid, and at least 13 volatiles in modern cultivated tomatoes [5]. The decline in tomato flavor quality has caused concern, and its improvement has become one of the hot spots of current research and breeding. It has been found that fruit weight and hardness have been preferentially selected in tomato breeding during long-term domestication and improvement, inadvertently resulting in the loss of alleles related to sugar and volatile synthesis [6,7]. Soluble sugars and organic acids provide the basic flavor for fruits and vegetables, while the characteristic aromas of volatiles are the main indicator that distinguish flavors among different fruits and vegetables [8,9].

The most important volatiles of tomatoes come from the green parts (stems, branches, leaves, sepals, and petioles), which provide the green leaf aroma. They are referred to as green leaf volatiles (GLVs) and include such compounds as n-hexanal, n-hexanol, (E)-2-hexenal, (E)-2-hexenol, (Z)-3-hexenal, (Z)-3-hexenol, (E)-3-hexenal, (E)-3-hexenol, and (Z)-3-hexenyl acetate. GLVs can transmit signals, induce the internal defense response of plants, attract predators to eat herbivorous insects, have some antibacterial effects, and enhance the adaptability of plants to adversity [10,11,12]. GLVs can be transferred to tomato fruits, increasing their grassy aroma and improving fruit freshness.

An analysis of the volatile contents and compositions of tomato fruits showed that fatty acid-derived volatiles accounted for more than half of the total volatiles, with C6 volatiles being especially higher. They are the main GLVs. Phenylpropane-derived volatiles account for one-third of the total volatiles, whereas the levels of carotenoid-derived volatiles and branched-chain amino acid-derived volatiles are much lower. The fatty acid-derived volatiles have grassy and fatty aromas; the phenylpropane-derived volatiles have irritation and floral aromas; the branched-chain amino acid-derived volatiles have a caramel aroma; and the carotenoid-derived volatiles have floral, fruity, sweet-like, and fatty aromas [13,14,15]. Carotenoid-derived volatiles are welcomed by consumers [16,17]. However, they appear to be relatively scarce in tomato fruits. Fortunately, since they have low threshold concentrations, some scientists speculate that increasing the levels of carotenoid-derived volatiles could significantly improve the tomato flavor quality [5,18].

CCO enzymes can cleave carotenoid at multiple unsaturated C=C double bond sites in a symmetric or asymmetric manner to produce abundant apocarotenoids and their carotenoid-derived volatiles. The CCOs in tomatoes mainly include seven carotenoid cleavage dioxygenases (CCDs) and six 9-cis-epoxy-carotenoid dioxygenases (NCEDs) [19,20]. CCD and NCED families have low homology and different substrate specificities [21].

A wide range of CCD enzyme can cleavage linear carotenoid (lycopene) at the C5=C6, C7=C8, or C9=C10 double bonds and cyclic carotenoid (β -carotene) at the C9=C10 double bond [22,23]. The products of CCD enzyme cleavage of the carotenoid include 5,6-epoxy-3-hydroxyl-β-ionone, geranylacetone, β-ionone, C14 dialdehyde, 3-hydroxyl-β-ionone, 6-methyl-5-heptene-2-ol, 6-methyl-5-heptene-2-one, 9-cis-neoxanthin, 9-cis-neoxanthin, α-ionone, pseudoionone, 4,9-dimethyldodecane-4,6,8-trienedialdehyde, and 3-hydroxy-α-ionone [24,25].

Hundreds of volatiles have been found in tomatoes, but only a few dozen of them play a role in flavor [24,26]. Therefore, the identification of the carotenoid-derived volatiles that contribute most to tomato flavor is critical for improving the flavor quality of tomato fruits. Hence, it is of great importance to screen out the key *SlCCD* genes involved in the synthesis of carotenoid-derived volatiles.

## 2. Materials and Methods

### 2.1. Plant Materials

The materials used in this study were the inbred lines CI1005 (*Solanum lycopersicum* var. *cerasiforme*) and TI4001 (*Solanum lycopersicum* var. *lycopersicum*), provided by the Tomato Biotechnology and Genetic Improvement Lab of Tomato of Northwest A&F University, Yangling Zone, China. CI1005 is rich in volatiles and has a strong aroma. TI4001 is very low in volatiles and has a light aroma.

### 2.2. Sampling Methods

In 2018, the tomato seeds were sown on January 18, 2018 and planted in plastic greenhouses on March 16. In 2019, the tomato seeds were sown on January 25 and planted on March 20 in the Yangling Zone, Shaanxi Province of China. Standard commercial production practices were carried out. Tomato fruits on the third inflorescence were picked at the mature green (BBCH-Scal 79~81), turning (BBCH-Scal 82~84), orange (BBCH-Scal 85~87), and red ripe (BBCH-Scal 88~89) stages, respectively [27]. Twenty fruits were selected at each stage with uniform sizes and colorations and with no signs of cracks, diseases, insects, or decay [28]. The sepals were removed from the tomato fruits that were then crushed into a homogenate using a FJ200-SH homogenizer (specimen model factory, Shanghai, China) and subjected to subsequent experiments. Three biological replicates were set up.

### 2.3. Determination on Volatile Contents of Tomato Fruits

Qualitative and quantitative analyses of the volatile compounds were conducted using references in the literature [18,29,30].

### 2.4. Sensory Evaluation on Aroma Quality of Tomato Fruits

First, the test was reviewed by the Ethics Committee of Northwest A&F University. Second, volunteers were informed in advance of the potential hazards of the sensory evaluation test. Third, fifty female and fifty male volunteers, including those native to Asia, Europe, and Africa, were recruited, and their ages ranged from 18 to 60 years old [31]. Fourth, the volunteers trained for two weeks with regard to knowledge of food flavor chemistry and skills of tomato flavor sensory evaluation by six food flavor specialists from Northwest A&F University. Fifth, five female and five male volunteers were selected using a triangulation test [32] to sniff and describe the aroma characteristics and intensities of different volatiles using an olfactory detector. These volatiles were all from the homogenate of tomato fruits with sepals removed. The aroma intensity of each volatile was divided into six levels as very strong (21~25 points), strong (16~20 points), medium (11~15 points), light (6~10 points), very light (1~5 points), and odorless (0 point) [33].

### 2.5. Tissue-Specific Expression of SlCCD Genes in Tomato

The nucleotide sequences of the *SlCCD* genes were downloaded from the NCBI website (https://www.ncbi.nlm.nih.gov/ (accessed on 13 January 2018). Primers for gene quantification were designed using Primer 5.0 software according to the principles of real-time quantitative primer design (Appendix A), and the *SlActin* gene was used as the internal reference gene. The total tomato RNA was extracted using the TRIzol method [34]. First, the genomic DNA was removed. Then, mRNA samples were reverse-transcribed into cDNA using an Evo M-MLV Reverse Transcription Kit II (Accurate Biotechnology Co., Ltd., Wuhan, China). The qRT-PCR reactions were performed in QuantStudio 5 (Life Technologies Co., Ltd., Waltham, MA, USA). The reaction system volume was 20 µL, including 10 µL of 2 × SYBR^®^ Green Pro Taq HS Premix, 200-ng cDNA, 0.2 µM each of the forward and reverse primers, 0.4-µM ROX Reference Dye, and the volume supplemented with 20 µL with RNase-free ddH_2_O. The reaction conditions were pre-denaturation at 95 °C for 3 min, denaturation at 95 °C for 10 s, and annealed at 60 °C for 1 min for 40 cycles. Three biological replicates were used for each sample. The relative expression of the genes was calculated according to the formula 2^−^^ΔΔCt^.

### 2.6. Cloning of SlCCD1A and SlCCD1B from Tomato Fruits

The total RNA was extracted from tomato fruits using the AG RNAex Pro kit. The procedure was described in the AG RNAex Pro RNA extraction kit instructions (Accurate Biotechnology Co., Ltd.). The mRNA was reverse-transcribed into cDNA. The forward and reverse primers (Appendix A) were designed from the 5′ UTR and 3′ UTR of *SlCCD1A* and *SlCCD1B* to ensure amplification of the complete CDS nucleotide sequences. PCR amplified the CDS sequences (1638 bp) of *SlCCD1A* and *SlCCD1B*. The reaction system volume was 50 µL, including a 25-µL 2 × high-fidelity enzyme, 400-ng cDNA, 0.5 µM each of the forward and reverse primers, and the volume supplemented with 50 µL of RNase-free ddH_2_O. The PCR reaction conditions were 98 °C pre-denaturation 58 s, denaturation at 98 °C for 10 s, annealing at 58 °C for 5 s, extension at 72 °C for 30 s for 35 cycles, and a final extension at 72 °C for 10 min, with storage at 4 °C. After PCR amplification was finished, the target bands were detected by 1% agarose gel electrophoresis. If the specific target band was detected, the target band was recovered with a gel recovery kit (Omega, Shanghai, China) and sent to Bioengineering Co., Ltd. (Shanghai, China) for sequencing. Then, sequence alignment was performed with DNAMAN software or by BLAST searching sequences on the NCBI website, ensuring that the sequences repeat rate was 100%.

### 2.7. Analysis of Nucleotide and Promoter Sequence and Transcription Factors of SlCCD1A and SlCCD1B

The CDS sequences, exons, and introns of *SlCCD1A* and *SlCCD1B* were analyzed. The open reading frames of *SlCCD1A* and *SlCCD1B* were found using ORF Finder software (http://www.bioinformatics.org/sms2/orf_find.html (accessed on 3 April 2018). The nucleotide sequences of *SlCCD1A* and *SlCCD1B* were BLAST-searched using BoxShade software (https://embnet.vital-it.ch/software/BOX_form.html (accessed on 3 April 2018). Their conserved structural domains were identified using MEME Suit Version 5.1.1 (http://meme-suite.org/tools/meme (accessed on 4 April 2018).

The promoters of *SlCCD1A* and *SlCCD1B* were analyzed. The promoter sequences of 2000-bp upstream and 100-bp downstream of the start codons were downloaded from the NCBI website and analyzed online using PlantCARE software (http://bioinformatics.psb.ugent.be/webtools/plantcare/html/ (accessed on 5 April 2018). The promoter cis-acting regulatory elements were analyzed online using PlantCARE software [35].

The transcription factors that could bind to promoters of *SlCCD1A* and *SlCCD1B* were predicted using the PROMO database (http://alggen.lsi.upc.es/cgi-bin/promo_v3/promo/promoinit.cgi?dirDB=TF_8.3 (accessed on 5 April 2018). Then, we performed optimized screening of the transcription factors from the JASPAR database (http://jaspar.genereg.net/collection/core/ (accessed on 10 April 2018).

### 2.8. Analysis of Amino Acid Sequences of SlCCD1A and SlCCD1B Proteins

The amino acids of the SlCCD1A and SlCCD1B proteins were analyzed for their amino acid composition, hydrophilicity and hydrophobicity, and physicochemical properties using the online tool ExPaSy (http://web.expasy.org/protparam/ (accessed on 10 April 2018). The transmembrane structures of the proteins were analyzed using the online tool TMpred (https://embnet.vital-it.ch/software/TMPRED_form.html (accessed on 12 April 2018)). The online tools SOPMA (https://npsa-prabi.ibcp.fr/cgi-bin/npsa_automat.pl?page=/NPSA/npsa_sopma.html (accessed on 12 April 2018) and PHYRE serve v2.0 (http://www.sbg.bio.ic.ac.uk/phyre2/html/page.cgi?id=index(accessed on 13 April 2018) were used to predict the secondary and tertiary structures of both the SlCCD1A and SlCCD1B proteins, respectively. The online tools PHYRE serve v2.0, InterPro (http://www.ebi.ac.uk/interpro/ (accessed on 15 April 2018), and the Multiple Expectation Maximization for Motif Elicitation (MEME) program (http://meme.nbcr.net/meme/ (accessed on 15 April 2018) were used to analyze the proteins for the conserved structural domains [36,37]. A phylogenetic tree was constructed using MEGA5.1 software with the neighbor-joining method and the Poisson correction mode. The gap was set to complete deletion, and the calibration parameter was set to bootstrap = 1000 [38]. Then, the sequences of the SlCCD1A and SlCCD1B proteins were submitted to the STRING online tool (https://string-db.org (accessed on 16 April 2018); their protein interaction networks were predicted and constructed with the parameters set to the default values.

### 2.9. SlCCD1A Recombinant Vector Construction

#### 2.9.1. The Overexpression Recombinant Vector Construction of *SlCCD1A*

For construction of the overexpression recombinant vector of *SlCCD1A*, we refer the reader to the relevant literature [39]. A cloned CDS fragment of *SlCCD1A* was ligated into the pMD18-T and left at 16 °C overnight (8~10 h). Five microliters of recombinant pMD18-T vector were transferred into *E. coli.* DHα5-competent cells, then placed in an ice bath for 30 min, heat-excited at 42 °C for 90 s, and placed on ice for 2 min. Nine hundred microliters of LB liquid medium were added. The competent cells were incubated at 37 °C, 200 rpm for 1 h, then centrifuged at 4000 rpm for 1 min. Part of the supernatant was discarded and left 100~150 µL of bacterial liquid, which was spread evenly on LB solid medium containing ampicillin (Amp). Petri dishes with bacterial liquid were incubated at 37 °C. After about 8 h, monoclonal bacteria appeared. We picked out 8~10 monoclonal bacterial spots with a gun tip and gently put them into 1 mL of LB liquid medium containing Amp, respectively. We sealed the centrifuge tube containing the monoclonal bacterial spots. The monoclonal bacteria were incubated at 37 °C, 200 rpm, until the concentration of the bacterium reached OD_600_ = 0.8. A bacteria solution of 1 mL was taken as the template for PCR. We screened the positive strains by electrophoresis. Culturing was continued for the positive strains and the plasmid extract. The *SlCCD1A* fragment was digested from the recombinant pMD18-T vector with endonuclease (Thermo Fisher, Waltham, MA, USA) from the recombinant pMD18-T vector at 37 °C for 16~20 h. The *SlCCD1A* fragment was detected by electrophoresis and sequencing by Bioengineering Co., Ltd. (Shanghai, China). The *SlCCD1A* fragment was ligated into the overexpression vector pVBG2307 with kanamycin (Kan) resistance using the same method.

#### 2.9.2. The RNAi Recombinant Vector Construction of *SlCCD1A*

The *SlCCD1A* sequence was subject to BLAST searching on the NCBI website to identify the conserved regions. The sequence of 266 bp in the non-conserved region was selected as the RNA interference (RNAi) fragment. The RNAi fragment was first ligated with the pMD18-T vector digested with endonuclease (Thermo Fisher, USA) from the recombinant pMD18-T vector at 37 °C for 16~20 h. Then, the RNAi fragment was ligated with the RNAi vector pCAMBIA1300 (Kan-resistant). This method was similar to the construction of the recombinant vector for *SlCCD1A* overexpression.

### 2.10. Genetic Transformation of SlCCD1A Recombinant Vectors

The recombinant vectors were transformed with *Agrobacterium tumefaciens* GV3101 by referring to the relevant literature [40].

The bacterial solution was turbid. If the target gene fragment was detected by electrophoresis, then the target gene had successfully been transferred into *Agrobacterium*.

Details of the infiltration solution of *Agrobacterium tumefaciens c*an be found in the relevant literature [39].

The specific steps for the culture of tomato seedlings can be found in Reference [41].

Callus induction: Take the cotyledons and hypocotyls of tomato seedlings (12 d), cut into 0.5 cm × 0.5 cm sizes, and incubate in the dark for 48 h. Immerse them in the GV3101 engineering solution containing the recombinant vector. Allow GV3101 to infest the cotyledons and hypocotyls with constant shaking for 10 min. Place the infested cotyledons and hypocotyls back into the preculture medium at 28 °C for 36 h in the dark. The explants are transferred into the co-medium and cultured until resistant buds are produced. Then, the explants are transferred to the screening medium. When the resistant buds grow to 2~4 cm, they are cut off and transferred to a rooting medium.

Seedling refining and transplanting: Start to refine the seedlings after 4 to 5 leaves have grown and roots have formed, and open the cap of the tissue culture bottle with a small opening in the culture chamber. Gradually open the vials after 2 to 3 d of incubation. After another 2 to 3 d, remove the seedlings slowly, wash the roots from the culture medium, and transplant them into a sterilized substrate covered with transparent plastic cups to retain water. Place the seedlings into the light incubator to slow down their growth and maintain a suitable temperature, humidity, and light.

Transfer to the greenhouse for planting after one week.

Identification of positive transgenic plants: The genomic DNA of tomato plant leaves was extracted using the CTAB method, PCR-amplified with vector fragments as forward primers and target gene fragments as reverse primers, and detected by gel electrophoresis. Samples with target bands were considered as positive transgenic plants [42].

### 2.11. SlCCD1A Expression of Transgenic Tomato Plants

The methods for the *SlCCD1A* expression of transgenic tomato plant detection and calculation are the same as in Section 2.5.

### 2.12. Volatile Contents and Evaluation of Aroma Quality of Transgenic Tomato Fruits

The volatile content of transgenic tomato fruits was determined as described in Section 2.3. The evaluation of the aroma quality of transgenic tomato fruits was determined as described in Section 2.4.

### 2.13. Data Statistics and Analysis

All data were saved in WPS Office 2019. The standard deviations (SDs) and coefficients of variations (CVs) were obtained by analysis with SPSS 22.0 software [43]. At least three biological replicates were performed for each measurement indicator. The data were standardized using a Z-score [44]. The significant differences among samples were performed by a one-way ANOVA (*p* < 0.05) after the homogeneity of variance test. Pearson correlation coefficients and a principal component analysis were obtained using SPSS 22.0 software. Heat maps were obtained by analysis with TB tools (v1.082) software.

## 3. Results

### 3.1. Tomato Fruit Volatiles Composition

The volatile contents from fatty acid, carotenoid, phenylpropane, and branched-chain amino acid were 4992.06, 1472.3, 1894.63, and 447.3 μg/kg, respectively, at the different maturity stages of the tomato fruits of both TI4001 and CI1005, which are shown in Figure 1.

In TI4001, the contents of fatty acid, carotenoid, phenylpropane, and branched-chain amino acid-derived volatiles were 883.22, 104.49, 655.45, and 84.86 μg/kg, respectively. The contents of fatty acid, phenylpropane, and branched-chain amino acid-derived volatiles significantly increased from the mature green stage to the turning stage and decreased from the turning stage to the red stage. The content of carotenoid-derived volatiles gradually decreased (by 165.5 μg/kg).

In CI1005, the contents of fatty acid, carotenoid, phenylpropane, and branched-chain amino acid-derived volatiles were 4992.06, 1472.3, 1894.63, and 447.3 μg/kg, respectively. The fatty acid and carotenoid-derived volatiles gradually increased with the fruit ripening. The phenylpropane-derived volatiles significantly decreased from the mature green stage to the turning stage (by 615.4 μg/kg) and increased from the turning stage to the red stage. The content of the branched-chain amino acid-derived volatiles continued to increase from the mature green stage to the orange stage and decreased slightly from the orange stage to the red stage (by 30 μg/kg).

### 3.2. Aroma Characteristics of Tomato Fruits

Fatty acid, carotenoid, phenylpropane, and branched-chain amino acid-derived volatiles have their own unique flavor characteristics (Figure 2). The contributions of different metabolic pathway-derived volatiles to tomato flavor according to the aroma evaluation scores are shown in Figure 2. The fatty acid-derived volatiles contributed more to grassy (13), fatty (11), mushroom-like (8), floral (4), fruity (4), and irritation (4) aromas. Carotenoid-derived volatiles contributed more to floral (10), fruity (7), sweet-like (7), and fatty (3) aromas. Phenylpropane-derived volatiles contributed more irritation (13), fruity (5), sweet-like (2), and floral (1) aromas. Branched-chain amino acid-derived volatiles contributed slightly sweet-like (three) and grassy (two) aromas. The fatty acid-derived volatiles contributed more to a grassy aroma at the mature green stage and more to a fatty aroma at the orange and the red stages. Carotenoid-derived volatiles contributed more to floral and sweet-like aromas at the turning, orange, and the red stages and to a fruity aroma at the orange and the red stages. The phenylpropane-derived volatiles contributed more to an irritating aroma at the mature green and the orange stages. The branched-chain amino acid-derived volatiles contributed less to the aromas. Thus, carotenoid-derived volatiles contributed more to the tomato aromas.

### 3.3. Tissue-Specific Expression of SlCCD in Tomatoes

The results for the tissue expression levels of seven *SlCCDs* in tomatoes are shown in Figure 3. The expression levels of *SlCCD1A* and *SlCCD1B* in tomato fruits, *SlCCD4A* in the petals, and *SlCCD7* and *SlCCD8* in the roots were significantly higher in both TI4001 and CI1005. In TI4001, the expression level of *SlCCD4B* in the fruit was significantly higher than in other tissues at the turning stage, and *SlCCD-LIKE* in calyx was higher than in other tissues. In CI1005, the expression level of *SlCCD4B* was the highest in the bud and higher in the petals and green-ripening fruits. *SlCCD-LIKE* was significantly higher in the petals than in other tissues.

Between the two accessions, the expression levels of *SlCCD1A*, *SlCCD1B*, *SlCCD4A*, *SlCCD4B*, and *SlCCD8* were higher in CI1005 than TI4001. The *SlCCD-LIKE* expression levels were significantly higher than TI4001 in the leaves, buds, and petals. In contrast, the *SlCCD7* expression levels in the roots and *SlCCD-LIKE* expression levels in the calyx, young fruit, and stem were significantly higher in TI4001 than those in CI1005.

The *SlCCD* expression varied at the different stages of tomato fruit development. *SlCCD4B* was the highest at the mature green stage. *SlCCD1B* and *SlCCD-LIKE* were the highest at the turning stage. *SlCCD1A* was the highest at the red ripe stage. The expression levels of *SlCCD4A*, *SlCCD7*, and *SlCCD8* decreased with fruit ripening. This indicates that *SlCCD4B* was mainly expressed at the early mature stage, *SlCCD1B* was mainly expressed at a late mature stage, and the expression levels of the other *SlCCDs* were nearly unaffected by the fruit-ripening process.

### 3.4. Bioinformatics Analysis of SlCCD1A and SlCCD1B in Tomatoes

#### 3.4.1. Sequence Analysis of t*SlCCD1A* and *SlCCD1B*

The sequencing results are 100% similar to *SlCCD1A* (ID:554393) and *SlCCD1B* (ID:544269) on the NCBI website, both at 1638 bp. *SlCCD1A* (locus name NC_015438.3) is located on chromosome 1 between the 82,060,579 and 82,071,211-cM intervals, whose nucleotide chains extend in the opposite direction to chromosome 1.

The CDS region of *SlCCD1A* consists of 1638 nucleotides, including 14 exons, 13 introns, and a complete open reading frame that can encode 545 amino acids. Homologous gene *SlCCD1B* is located upstream of *SlCCD1A*. The similarity was 83.15% between *SlCCD1A* and *SlCCD1B* in the CDS nucleotide sequence. *SlCCD1B* is located on chromosome 1 between the 82,085,914 and 82,073,038-cM interval at locus NC_015438. The CDS region of *SlCCD1B* comprises 1638 nucleotides, including 14 exons, 14 introns, and a complete open reading frame that can encode 546 amino acids.

#### 3.4.2. The Promoter Analysis of *SlCCD1A* and *SlCCD1B* in Tomatoes

The promoter cis-acting regulatory elements of *SlCCD1A* and *SlCCD1B* are shown in Appendix A. The TATA box is the most frequently occurring and belongs to the 30-bp core promoter element upstream of the transcription start position. The transcription factors shared by *SlCCD1A* and *SlCCD1B* include the G box (responsive to light signal); AE box, chs-CMA2a, and TCT motif (partial module responding to light signal); Box 4 (partial conserved DNA module responding to light signal); circadian (circadian rhythm regulation); ABRE (responding to an ABA signal); and CAAT box (initiating and enhancing transcription). The regulatory elements specific to *SlCCD1A* are the O_2_ site (zeaxanthin metabolism) and TGACG motif and CGTCA motif (in response to MeJA signaling). The regulatory elements specific to *SlCCD1B* are MRE (involved in the light corresponding to the MYB-binding site), GATA motif and LAMP element (module in partial response to a light signal), GARE motif (in response to a GA signal), CAT box (involved in phloem expression), ARE (required for anaerobic induction), MBS (MYB-binding site involved in drought induction), ARE (maximal receptor for mediating activator), and HD-Zip 3 (protein-binding site). It is suggested that the expressions of *SlCCD1A* and *SlCCD1B* are affected by light, phytohormones, and abiotic stresses.

#### 3.4.3. Analysis of the *SlCCD1A* and *SlCCD1B* Transcription Factors in Tomatoes

Based on interactions with the promoter sequence of *SlCCD1A*, 86 transcription factors were screened in the PROMO database according to a tolerance of 0, and 11 transcription factors were then found in the JASPAR database according to a relative profile score threshold of >85% (Table 1). Using the same approach, 14 transcription factors interacting with the promoter of *SlCCD1B* were found. Ten transcription factors may interact with both the *SlCCD1A* and *SlCCD1B* promoters. Dof2, YY1, hb, and MNB1A are zinc-finger transcription factors. Nkx2-5, HNF1A, and PAX are conserved factors. CEBPA is a leucine-rich zipper factor, MYB is a tryptophan-rich factor, and TBP is a TATA-binding protein. In addition, the Gata1 (zinc finger) transcription factor only interacts with the *SlCCD1A* promoter. The GATA2 and PBF (zinc finger), E2F1 (tryptophan-rich), and IRF2 (forkhead/winged-helix) transcription factors only interact with the *SlCCD1B* promoter. It is suggested that *SlCCD1A* and *SlCCD1B* expressions are regulated by zinc finger, leucine-rich, and tryptophan transcription factors.

#### 3.4.4. Structural Analysis of the SlCCD1A and SlCCD1B Proteins in Tomatoes

The SlCCD1A protein has a molecular formula of C_2763_H_4285_N_731_O_795S23_, molecular weight of 61.20 kDa, isoelectric point of 6.14, and consists of 545 amino acids. It contains 20 types of amino acids, with high numbers of leucine (8.40%), glycine (8.30%), and valine (8.30%). The number of cysteine (0.90%) and tryptophan (0.70%) is low. The number of negatively charged amino acid residues (Asp + Glu) is 71, and the number of positively charged amino acid residues (Arg + Lys) is 64. The absorbance value at OD_280_ nm is 0.850, and the extinction coefficient is 52,050/M/cm. The in vitro half-life is 10~30 h. The value of the instability index (II) is 31.05, which indicates a stable protein. The lipid index value is 81.54. The mean water solubility is −0.272, corresponding to a hydrophobic protein. The SlCCD1A protein has a transmembrane structure from the cytoplasm to the extracellular at position 140~158 aa, predicted by TMpred software. The SlCCD1A protein contains six tandem transmembrane structures at 13~34 aa (S1), 71~98 aa (S2), 107~128 aa (S3), 136~163 aa (S4), 193~210 aa (S5), and 245~264 aa (S6), predicted by PHYRE 2 software. S1, S3, and S5 are extracellular to cytoplasmic, and S2, S4, and S6 are cytoplasmic to extracellular, with both the N- and C-termini outside the cell. A protein secondary structure analysis revealed that the SlCCD1A protein accounts for 51.74% of the irregular coiling, 24.59% of the extended chain, 18.17% of the α-helix, and 5.50% of the β-turn.

The amino acid sequence similarity is 83.15% between SlCCD1B and SlCCD1A. The amino acid composition, physicochemical properties, secondary structure, and tertiary structure of SlCCD1B are more similar to those of the SlCCD1A protein. The SlCCD1B protein has a molecular formula of C_2771_H_4270_N_720_O_800_S_22_, molecular weight of 61.18 kDa, and isoelectric point of 5.65. The 545 amino acids consisted of 20 types of amino acids. The number of glycines, valines, and leucines are high, while the number of tryptophans and cysteines is low. The absorbance value is 0.914 at OD_280_ nm. The extinction coefficient is 55,935/M/cm. The in vitro half-life is 10~30 h. It is a hydrophobic stable protein. There is a transmembrane structure from the cytoplasm to the extracellular at position 145~165 aa. The secondary structure is 53.58% irregularly coiled, 24.22% an extended chain, 16.51% an α-helix, and 5.69% a β-turn.

The tertiary structures of the SlCCD1A and SlCCD1B proteins are shown in Figure 4. The protein tertiary structure resembles a β-propeller, with each blade consisting of four to five reverse-parallel β-turns. The α-helix is at the top of the β-propeller surface, and irregular curls wind the propeller up. The structural similarity with viviparous 14 (VP 14) found in the maize endosperm is up to 92%. It is functionally annotated as having oxidoreductase activity, acting on a single donor through the addition of molecular oxygen, with 9-cis-epoxycarotenoid dioxygenase activity.

#### 3.4.5. Conserved Structural Domains and Phylogenetic Analysis on the CCD1 Protein in Dicotyledonous Plants

There are 48 *CCD1* homologs included on the NCBI website; among which, 20 belong to dicotyledonous plants. The amino acid sequence comparison of the CCD1 proteins in dicotyledonous plants revealed that the similarity is as high as 84.74%. The conserved domains of the protein predict that the *CCD1* belong to the PLN02491 subfamily of the retinal pigment epithelium (RPE) 65 (cl10080) supergene family. In plants, RPE65 is associated with neoxanthin cleavage enzymes. Oxidative cleavage is performed by adding two oxygen atoms to a single carotenoid molecule. The top 10 conserved domains (motifs) with high maximum likelihood values are shown in Figure 5. The logos of the most conserved domains with higher likelihood values are also given. The letter size indicates the frequency of amino acid occurrence.

The amino acid sequence similarity of the seven SlCCD members is only 38.63%, although they belong to the same RPE65 superfamily. There is only one conserved domain, EDDG. The SlCCD1A, SlCCD1B, SlCCD4A, and SlCCD4B proteins belong to the PLN02491 subfamily; the SlCCD7 protein belongs to the PLN02969 subfamily; and the SlCCD8 and SlCCD-LIKE proteins belong to the pfam03055 subfamily. By contrast, an amino acid sequence analysis of the seven SlCCD members revealed that their similarities were only 38.63%, with only one conserved domain, EDDG.

The conserved domains of SlCCD1A are shown in Figure 6, which gives the top 10 conserved domains, with a high value indicating a greater likelihood. This can represent the distribution of the conserved domains on the SlCCD1A amino acid sequence and the amino acid sequences of the conserved domains.

A phylogenetic analysis of 20 CCD1 amino acid sequences of dicotyledons (Figure 7) showed they can be classified into four categories (from bottom to top) based on their phylogenetic relationships. The first category includes carrot (DcCCD1 and DcCCD1-like), bitter melon (McCCD1), *Arabidopsis thaliana* (AtCCD1), warm mandarin (CituCCD1), tomato (SlCCD1A and SlCCD1B), and saffron (CsCCD1). This indicates that the SlCCD1A and SlCCD1B proteins are more conserved in evolution and have similar functions. The second group includes sunflowers (HaCCD1-like), melons (CmCCD1), and laurels (OfCCD1). The third group includes small-grain coffee (CaCCD1) and medium-grain coffee (CcCCD1). The fourth group includes common tobacco (NtCCD1-like1 and NtCCD1-like2), petunias (PhCCD1), grapes (VvCCD1), *Tribulus alfalfa* (MtCCD1), woody plant cassava (MeCCD1), and Turk’s roses (RdCCD1).

#### 3.4.6. Regulatory Network Analysis of the SlCCD1A and SlCCD1B Proteins

The regulatory networks for the SlCCD1A and SlCCD1B proteins are shown in Figure 8. The SlCCD1A protein is closely linked with proteins of the carotenoid metabolic pathway, especially deoxy-D-xylulose-5-phosphate reductoisomerase (DXR), octahydro lycopene synthase (PSY), octahydro lycopene dehydrogenase (pds), z-carotene desaturase (zds), carotenoid synthase (CRTISO), and carotenoid oxidases such as lipoxygenase (LoxC).

By contrast, there is no deoxy-D-xylulose-5-phosphate reductoisomerase (DXR) and octahydro lycopene synthase (PSY) in the SlCCD1B protein regulatory network. Instead, amino acid decarboxylase 1B (AADC1B), β-glucosidase (100191128), ethanol dehydrogenase (yfe37), and glutathione S-transferase/peroxidase (BI-GST/GPX) are present. This indicates that the SlCCD1A protein may be more closely related to carotenoid metabolism than the SlCCD1B protein. The SlCCD1B protein is closely related to amino acid metabolism, sugar metabolism, and fatty acid metabolism.

### 3.5. Effect of SlCCD1A on Volatiles and Aroma in Tomato Fruits

#### 3.5.1. Analysis of SlCCD1A Expression in Transgenic Overexpressing Lines of Tomato Fruits

The expression levels of *S**lCCD1A* in OE lines of tomato fruits are shown in Figure 9. A total of six OE lines (OE-3, OE-5, OE-6, OE-8, OE-9, and OE-11) were obtained from CI1005 tomatoes. The expression levels of *S**lCCD1A* in the OE-3, OE-11, and OE-8 lines were significantly higher by 3.5, 2.98, and 2.96 times, respectively, than in WT tomatoes (transferred into PVBG2307), which were 6.18 times that of WT at the red stage. The expression levels of *S**lCCD1A* in the OE-5, OE-6, and OE-9 lines were not significantly different from that of the WT.

A total of five OE lines (OE-2, OE-3, OE-6, OE-7, and OE-9) were obtained from TI4001 tomatoes, and the expression levels of *S**lCCD1A* in the OE-2, OE-6, and OE-3 lines were significantly higher by 4.47, 4.02, and 3.32 times, respectively, than that of WT tomatoes (transferred into PVBG2307). The expression levels of *S**lCCD1A* in the OE-7 and OE-9 lines were not significantly different from that of the WT.

#### 3.5.2. Analysis of *SlCCD1A* Expression in RNAi Transgenic Tomato Fruits

Five RNAi lines (RANi-2, RNAi-4, RNAi-5, RNAi-7, and RNAi-10) were obtained from CI1005 tomatoes, and the expressions levels of *SlCCD1A* in the fruits of each line are shown in Figure 10a. Among them, the expression levels of *SlCCD1A* in the RNAi-2, RNAi-7, and RNAi-10 fruits were significantly lower than in the WT (transferred into the pCAMBIA1300 empty vector), which were 24.73, 26.45, and 31.01% of the WT expression levels. It decreased the most obviously at the turning stage, only 18.67% of the WT. The expression levels of *SlCCD1A* in the RNAi-4 and RNAi-5 lines were lower than that of the WT, but the differences were insignificant.

A total of five RNAi lines (RANi-1, RNAi-3, RNAi-4, RNAi-5, and RNAi-7) were obtained from TI4001 tomatoes, and the expression levels of *SlCCD1A* in the fruits of each line are shown in Figure 10b. The expression levels of *SlCCD1A* in RNAi-4, RNAi-1, and RNAi-7 fruits were significantly lower 23.32, 23.44, and 32.11%, respectively, than in the WT (transferred into the pCAMBIA1300 empty vector). It decreased the most significantly at the red stage, only 21.45% of the WT. The expression levels of *SlCCD1A* in the RNAi-3 and RNAi-5 lines were lower than that of the WT, but the difference was not significant.

### 3.6. Effects of SlCCD1A on Volatiles in Tomato Fruits

#### 3.6.1. Overexpression of SlCCD1A on Volatile Accumulation in Tomato Fruits

The contents of the carotenoid-derived volatiles were significantly increased in the CI1005 OE lines of *SlCCD1A* (Table 2). Compared to the WT, (E)-citral, β-cyclocitral, 6-methyl-5-heptene-2-ol, geranylacetone, 6-methyl-5-heptene-2-one, (E)-a-ionone, β-ionone, geraniol, 3,7-dimethyl-6-octene-1-ol, and neral were found to be significantly increased in the fruits of three OE lines (OE-3, OE-8, and OE-11) of CI1005. The contents of acrylacetaldehyde, (E)-farnesal, pseudoionone, hexahydropseudovionone, and farnesyl acetone were increased.

Only the contents of geranylacetone, farnesyl acetone, and 6-methyl-5-heptene-2-ol in OE lines at the mature green stage were significantly higher than the WT, which were increased by 4.87, 3.1, and 2.2 μg/kg, respectively. (E)-Farnesal and 3,7-dimethyl-6-octene-1-ol were only detected in OE lines at the turning stage. Hexahydropseudovionone was only detected in OE lines at the orange stage. The contents of the other carotenoid-derived volatiles were significantly higher in OE lines than the WT. The content of each carotenoid-derived volatile in the OE lines was significantly higher than those in the WT at the red stage.

The *SlCCD1A* OE significantly increased the contents of the carotenoid-derived volatiles in TI4001 fruits (Table 3), especially the contents of 6-methyl-5-heptene-2-one, geranylacetone, and (E)-á-ionone. The contents of 6-methyl-5-heptene-2-one and geranylacetone in the OE-2, OE-3, and OE-6 OE lines of TI4001 were significantly higher than those in the WT at the turning, orange, and the red stages. The (E)-á-ionone content was significantly higher than that in the WT at the orange stage. The contents of β-ionone, geraniol, neral, and pseudoionone were higher than those in the WT, but the difference was insignificant.

#### 3.6.2. Effect of SlCCD1A RNAi on the Volatile Contents in Tomato Fruits

The *SlCCD1A* RNAi significantly reduced the contents of carotenoid-derived volatiles in CI1005 fruits (Table 4). Compared with the WT, the contents of the carotenoid-derived volatiles in three RNAi lines, RNAi-2, RNAi-7, and RNAi-10, were significantly lower. Geranylacetone, farnil acetone, 6-methyl-5-heptene-2-ol, and neral were not detected in the RNAi lines at the mature green stage. The 6-methyl-5-heptene-2-one content was significantly lower than that of the WT (35.15%) at the turning stage. Volatiles 6-methyl-5-heptene-2-ol, 6-methyl-5-heptene-2-ol, (E)-farnesal, and acrylacetaldehyde were not detected in the RNAi lines. The contents of 6-methyl-5-heptene-2-one, geranylacetone, (E)-á-ionone, and geraniol were significantly decreased at the orange stage. The contents of β-cyclocitral, 6-methyl-5-heptene-2-one, geranylacetone, (E)-á-ionone, and farnesyl acetone in the RNAi lines were significantly decreased. β-cyclocitral, 6-methyl-5-heptene-2-one, geranylacetone, (E)-á-ionone, and farnesyl acetone were not detected. The contents of (E)-citral, 6-methyl-5-heptene-2-one, geranylacetone, (E)-á-ionone, geraniol, pseudoionone, and acetone in the RNAi lines at the red stage were significantly decreased compared to those in the WT.

The *SlCCD1A* RNAi significantly reduced the contents of the carotenoid-derived volatiles in TI4001 fruits (Figure 11). The contents of 6-methyl-5-heptene-2-one and geranylacetone in three RNAi lines, RNAi-1, RNAi-4, and RNAi-7, of TI4001 were significantly lower than the WT at the turning, orange, and the red stages, while the differences of the other carotenoid-derived volatiles were not significant between the RNAi lines and the WT.

### 3.7. Effect of SlCCD1A on Aroma in Tomato Fruits

#### 3.7.1. OE of *SlCCD1A* on the Aroma in Tomato Fruits

The expression level of *SlCCD1A* mainly affected the floral, fruity, sweet-like, and fatty aromas of CI1005 fruits (Figure 12a). Compared with the WT, the fruity, floral, sweet-like, and fatty aromas of the three OE lines (OE-3, OE-8, and OE-11) of CI1005 were significantly enhanced. The fruity aroma scores of the CI1005 fruits increased four, eight, four, and five points at the mature green, turning, orange, and red ripe stages, respectively. The floral aroma scores increased five, five, six, and five points at the mature green, turning, orange, and red ripe stages. The sweet-like scores increased one, three, two, and three points at the mature green, turning, orange, and red ripe stages. The fatty aroma scores increased one, one, and two points at the turning, orange, and red ripe stages, respectively.

The expression level of *SlCCD1A* mainly affected the floral, fruity, and sweet-like aromas of TI4001 fruits (Figure 12b). The fruity aroma of three OE lines (OE-2, OE-3, and OE-6) increased by three, three, and two points; the floral aroma increased by six, four, and two points; and the sweet-like aroma increased by two, three, and two points at the turning, orange, and the red stages, respectively. Other aroma scores were not significantly different.

#### 3.7.2. RNAi of *SlCCD1A* on the Aromas in Tomato Fruits

The *SlCCD1A* RNAi could significantly reduce the aroma of CI1005 fruits (Figure 13a). Compared with the WT, the fruity, floral, sweet-like, and fatty aromas of three RNAi lines (RNAi-2, RNAi-7, and RNAi-10) were significantly reduced. Compared with the WT, the fruity aroma reduced by two, three, three, and two points; the floral aroma reduced by two, five, two, and two points; the sweet-like aroma reduced by two, three, two, and two points, and the fatty aroma decreased by zero, one, two, and three points at the mature green, turning, orange, and red stages, respectively.

The *SlCCD1A* RNAi could significantly reduce the aroma of TI4001 fruits (Figure 13b). Compared with the WT, the scores of the fruity, floral fragrance, and sweet-like aromas in the three RNAi lines (RNAi-1, RNAi-4, and RNAi-7) of TI4001 decreased by two, two, and one points at the turning stage, respectively. The floral aroma score decreased by one point at the orange stage. The other aroma scores were not significantly different.

### 3.8. Effects of SlCCD1A on Carotenoid Contents in Tomato Fruits

#### 3.8.1. OE of SlCCD1A on Carotenoid Contents in Tomato Fruits

The *SlCCD1A* OE significantly reduced the carotenoid contents of tomato fruits (Figure 14). Compared with the WT, the content of carotenoid in OE tomato fruits decreased significantly, especially after the turning stage. The contents of lutein, zeaxanthin, α-carotene, β-carotene, and lycopene in the fruits of three CI1005 OE lines (OE-3, OE-8, and OE-11) were only 2.75, 7.14, 10.96, 77.39, and 78.36% those of the WT, respectively. The contents of zeaxanthin, lutein, α-carotene, and β-carotene in the fruits of three OE lines (OE-2, OE-3, and OE-6) of TI4001 were only 14.02, 15.12, 65.83, and 78.53% those of the WT, respectively.

#### 3.8.2. RNAi of SlCCD1A on Carotenoid Contents in Tomato Fruits

The *SlCCD1A* RNAi could significantly increase the content of carotenoid in tomato fruits. Compared with the WT, the carotenoid contents of the RNAi lines were significantly higher than in the WT (Figure 14). The contents of zeaxanthin, α-carotene, lutein, β-carotene, and lycopene in three RNAi lines (RNAi-2, RNAi-7, and RNAi-10) of CI1005 were 8.04, 5.72, 4.13, 2.37, and 1.33 times those of the WT, respectively. The contents of α-carotene, zeaxanthin, lutein, and β-carotene in the fruits of three RNAi lines (RNAi-1, RNAi-4, and RNAi-7) of TI4001 were 4.20, 2.44, 2.15, and 1.81 times those of the WT, respectively. More lycopene was detected, with an average content of 0.23 mg/100 g.

## 4. Discussion

With improvements in the living standards, the sensory evaluation was found to be positively correlated with the tomato wholesale price [45]. The contribution of volatiles to tomato aroma has received increasing attention as research progresses [46]. In this study, a total of 72 volatiles were detected during the ripening of CI1005 tomato fruits, of which 44, 53, 57, and 55 volatiles were detected at the mature green, turning, orange, and the red stages, respectively. The total volatile contents were 4244.45, 7126.05, 10,555.15, and 13,299.5 μg/kg at these four stages. The volatile contents of the fatty acid, carotenoid, phenylpropane, and branched-chain amino acid derivatives were 4992.06, 1472.3, 1894.63, and 447.3 μg/kg, respectively, which accounted for 54.01, 14.18, 26.26, and 5.55% of the total volatile contents, respectively. The contents of the fatty acid and carotenoid-derived volatiles increased gradually with the fruit ripening. The content of the phenylpropane-derived volatiles decreased significantly from the mature green stage to the turning stage (by 615.4 μg/kg) and continued to increase from the turning stage to the red stage. The content of the branched-chain amino acid-derived volatiles continued to increase from the mature green stage to the orange stage and decreased from the orange stage to the red stage (by 30 μg/kg). The fatty acid-derived volatiles were mainly C6~C10 alcohols and aldehydes. The abundant C6 volatiles had a grassy aroma, the C8 volatiles had a mushroom-like aroma, and the C9 to C10 volatiles had a fatty aroma. Phenylpropane-derived volatiles (phenols and guaiacol) have phenolic and herbal aromas, and 2-phenylethanol has a floral aroma. Branched-chain amino acid-derived volatiles (2-methyl-butanol, 2-methyl-butyraldehyde, 3-methyl-butanol, and 3-methyl-butanal) have a malty aroma, and 2-isobutylthiazole has the grassy aroma of tomato stems. Carotenoid-derived volatiles have a floral aroma, such as 6-methyl-5-hepten-2-one, geranylacetone, and β-cyclic citral, which have floral, fruity, sweet-like, and fatty aromas that are preferred by consumers [47]. The carotenoid-derived volatile contents are very low. However, they may be the main factor affecting tomato aromas. Carotenoid-derived volatiles are popular for their aromas and significantly contribute to tomato flavors [5]. However, their contents are decreased in cultivated tomato species. The threshold analysis found that a trace increase in the contents of these volatiles could improve the tomato aromas [48]. The volatiles derived from carotenoid are mainly geranylacetone and 6-methyl-5-hepten-2-one. Although the contents of the carotenoid-derived volatiles are low, the olfactory threshold concentrations of these volatiles are also low. Only trace increases in the carotenoid-derived volatiles could improve the tomato flavor quality [49].

Carotenoids have diverse bioactive and chemical properties [50] and can be cleaved by *SlCCD*. In this study, we found that *SlCCD1A* and *SlCCD1B* were more significantly highly expressed in tomato fruits than in other tissues. The accumulation of carotenoid and their derived volatiles showed the same trend. The *SlCCD1A* and *SlCCD1B* expression levels were significantly and positively correlated with carotenoid and their derived volatiles. Simkin [48] found that, during fruit ripening, the *LeCCD1A* expression gradually decreased from the mature green stage to the orange stage and rapidly increased from the orange stage to the red stage. The *LeCCD1A* expression level reached a maximum at the red stage. By contrast, the trend of the *LeCCD1B* expression was opposite to that of the *LeCCD1A*. the *LeCCD1B* expression level reached a maximum at the orange stage. Ilg [25] showed that *LeCCD1B* was highly expressed in ripe tomato fruits, and the cleavage product contributed the most to the fruit flavor. Wei [51] concluded that *SlCCD1A* expression was highest at the mature green stage, rapidly decreased at the breaker stage, and slightly increased at the pink stage. The *SlCCD1B* and *SlCCD1A* expression trends were the opposite. The lowest expression appeared at the pink stage, while the highest expression appeared at the red stage. There may be differences in the *SlCCDA* and *SlCCD1B* expression trends among different tomato inbred lines.

*SlCCDA* and *SlCCD1B* are tandemly replicated [52]. *SlCCD1A* is located downstream of *SlCCD1B*. Both contain 1638 nucleotides in the CDS region. The sequence similarity is 83.15% between *SlCCDA* and *SlCCD1B*, containing a complete open reading frame that encodes 545 amino acids. A sequence analysis of the *SlCCD1A* and *SlCCD1B* promoters revealed that both promoter regions contain cis-acting regulatory elements that respond to light, abscisic acid signals, and circadian rhythms. The *SlCCD1B* promoter has elements that respond to gibberellin, cytokinin, and drought stress. Previously, the cis-acting elements involved in light response, hormone regulation, and abiotic and biotic stresses were identified in the promoter sequence of *SlCCD* [51]. In addition, the carotenoid content is also regulated by light, ethylene, and growth hormones [52]. The exogenous spraying of growth hormones and methyl jasmonate increased the isoprene volatile contents [26,53], indirectly demonstrating that growth hormones and methyl jasmonate positively regulate *SlCCD1A* expression. The transcription factors that can interact with the *SlCCD1A* and *SlCCD1B* promoters are the zinc-finger transcription factors, conserved factors, leucine-rich zipper factors, tryptophan-rich factors, and TATA-binding proteins. The Osmanthus *OfCCD1* promoter is regulated by the MYB and WAKY3 transcription factors [54].

A comparison of the dicotyledonous CCD1 amino acid sequences revealed an 84.74% similarity. A protein conserved structure analysis revealed that *CCD1* genes belong to the retinal pigment epithelial REP65 supergene family. RPE65 is a retinal pigment epithelial membrane protein that is abundantly expressed in the retinal pigment epithelium and binds to plasma retinal-binding proteins. In plants, RPE65 is associated with neoxanthin cleavage enzymes. Oxidative cleavage is performed by adding two oxygen atoms to a single carotenoid molecule [55]. The first carotenoid oxygenating CCO-VP14 was discovered in maize endosperm in 1997 [56]. Sequence homology studies with VP14 revealed that carotenoid oxygenases (CCOs) in plants mainly include two families of carotenoid cleavage dioxygenases (CCDs) and 9-cis-epoxy-carotenoid dioxygenases (NCEDs) [57]. NCED enzymes have a high homology with VP14, and CCD enzymes are functionally differentiated and can specifically catalyze more substrates [58,59]. CCO enzymes contain a conserved seven-blade formation of a β-propeller that shrouds four histidine motifs and the RPE65 structural domain. Each blade consists of four or five inversely parallel β-folds with 11 N-terminal residues freely aligned. An α-helix domain is formed by four inserted α-helices at the top of the β-propeller, contributing to the substrate diversity and functional diversification of the CCO enzymes. The active site structure of the CCO enzymes differs from other non-heme iron-dependent enzymes in that Fe^2+^ is bound at the distal end center of the β-propeller, surrounded by four histidines binding two water molecules to form the reactive center, which can regulate the substrate-binding efficiency [60,61]. Catalyzed by a single ferrous cofactor and O_2_ molecule, the CCO enzymes extract all 4-valent-reducing equivalents from the major substrate, add two O atoms, and cleave the electron-rich isoprene hydrocarbon chain symmetrically or asymmetrically to produce apocarotenoid, volatiles, and nonvolatiles [59].

CCO enzymes are specific non-heme iron-dependent enzymes with low amino acid sequence similarities and high-functional differentiation among family members [62], e.g., the seven members of the SlCCD family have only 38.63% amino acid sequence similarity and belong to the RPE65 superfamilies PLN02491 and PLN02969 and the pfam03055 subfamily. By contrast, the amino acid sequence similarity of 20 CCD proteins in dicotyledons was as high as 84.74%, all of which belonged to the PLN02491 subfamily. Phylogeny can be classified into four categories, and tomatoes SlCCD1A and SlCCD1B, which are more closely related to CCD1 petunias and common tobacco in the same family of *Solanaceae*, are clustered into one category.

A protein regulation network analysis showed that the SlCCD1A protein was closely related to DXR, PSY, pds, zds, and CRTISO, key enzymes of carotenoid synthesis and carotenoid oxidase (LoxC), while, in the SlCCD1B protein regulation network, DXR, PSY1, and PSY2, key enzymes of carotenoid synthesis, were not found, but AADC1B and BI-G, which are involved in amino acid metabolism, were found. *SlCCD1A* and *SlCCD1B* have tandemly replicated genes that have undergone purifying selection during diverse evolution and often differ in substrate selection, tissue localization, and expression regulation [44]. It was shown that SlCCD1A is more closely related to carotenoid metabolism than SlCCD1B. SlCCD1A is expected to be a key target gene for tomato aroma enhancement. Similarly, a transcriptional analysis of CsCCDs in tomato flowers and young fruits, together with structural and motif analyses of their proteins, revealed that *LeCCD1* expression has an important role in the synthesizing of carotenoid-derived volatiles [48].

We found the contents of lutein, zeaxanthin, and α-carotene in fruits to be significantly decreased in OE lines, with only 7.7, 7.8, and 28.9% of the WT, respectively. The contents of carotenoid-derived volatiles (E)-á-ionone, geranylacetone, 6-methyl-5-heptene-2-ol, and β-ionone increased 2.43, 2.34, 2.33, and 1.70 times, respectively, compared to the WT. More β-ionone and (Z)-citral *SlCCD1A* OE had no significant effect on the expressions of the other *SlCCDs*. In lines of antisense expressing *LeCCD1A* and *LeCCD1B*, the mRNA levels were decreased by 87~93%, and the contents of the carotenoid-derived volatiles (β-ionone and germanylacetone) were decreased by only 50 and 60%, respectively [48]. In this study, it was found that *SlCCD1A* had significant effects on the floral, fruity, fatty, and sweet-like aromas of tomato fruits. The scores of the fruity, floral, sweet-like, and fatty aromas of the CI1005 OE lines were increased by five, five, two, and one points, respectively, while the scores of the floral, fruity, and sweet-like aromas of the TI4001 OE lines were increased by four, three, and two points, respectively.

It was found that the expression levels of *SlCCD1A* in tomato RNAi fruits was only 22.59% that of the WT, and the contents of α-carotene, lutein, β-carotene, zeaxanthin, and lycopene were 5.35, 4.28, 2.32, 1.90, and 1.52 times that of the WT, respectively. The contents of 6-methyl-5-heptene-2-one, (E)-á-ionone, β-ionone, and (Z)-citral were only 23.5, 24.1, 33.4, and 41.2% those of the WT. This suggests that carotenoid-derived volatiles also are post-transcriptionally regulated or are related to other enzymes [24,63]. Compared with the WT, lycopene was detected in the TI4001 RNAi lines, but 6-methyl-5-heptene-2-one was not detected. *SlCCD1A* RNAi significantly reduced the expression of *SlCCD1B* in CI1005 during the turning and the red stages but did not significantly change the expression levels of the other *SlCCDs*. The scores of the fruity, floral, sweet-like, and fatty aromas in the CI1005 RNAi lines decreased by two, three, two, and one points, respectively, while the scores of the floral, fruity, and sweet-like aromas in the TI4001 RNAi lines increased by two, two, and one points at the turning stage. Due to the low odor threshold, only trace amounts of the carotenoid-derived volatiles can affect tomato aromas [48]. It was speculated that increasing the *SlCCD1A* expression is an effective way to increase the tomato aroma intensity.

It was found that the expression level of *SlCCD1A* significantly affected the contents of lutein, zeaxanthin, and α-carotene but had no effect on the contents of lycopene and β-carotene. On the one hand, the lycopene content in tomato fruits is as high as dozens of times that of zeaxanthin and lutein. Although the amount of lycopene cleaved by *SlCCD1A* is relatively large, the reduction percentage is small. On the other hand, the SlCCD1A enzyme may mainly cleave cyclic lutein, zeaxanthin, and α-carotene, producing α-ionone and β-ionone, but with less cleavage than linear lycopene. *Escherichia coli* showed that the recombinant FcCCD1A enzyme of *Ficus carica* L. was specific for the C9=C10 (C9′=C10′) double bond of cyclic carotenoid, producing α-ionone and β-ionone, while FcCCD1B cleaved the acyclic parts of lycopene and β-carotene to produce 6-methyl-5-heptene-2-one [64].

## 5. Conclusions

The volatile contents from fatty acid, carotenoid, phenylpropane, and branched-chain amino acid in tomato fruits were 4992.06, 1472.3, 1894.63, and 447.3 μg/kg, respectively, with percentage contents of 54.01, 14.18, 26.26, and 5.55%, respectively, on the whole.

The volatiles derived from fatty acid contributed to grassy, fatty, mushroom-like, floral, fruity, and fatty aromas. The carotenoid-derived and phenylpropane-derived volatiles contributed to floral, fruity, sweet-like, and fatty aromas. The branched-chain amino acid-derived volatiles contributed to sweet-like and grassy aromas.

The expression levels of *SlCCD1A* and *SlCCD1B* were significantly higher in tomato fruits than in other tissues and consistent with the trend of carotenoid-derived volatile contents.

Both *SlCCD1A* and *SlCCD1B* have cis-acting elements for light, hormone, and stress responses. The *SlCCD1A* promoter contains the cis-acting regulatory element O_2_ site in response to zeaxanthin metabolism, and the SlCCD1A and SlCCD1B proteins may regulate seven and four key enzymes for carotenoid metabolism, respectively.

The carotenoid-derived volatiles were significantly increased in the *SlCCD1A* OE lines, and the floral, fruity, and sweet-like aromas were significantly enhanced, while the carotenoid-derived volatiles were significantly decreased in the RNAi lines, and the floral, fruity, and sweet-like aromas were significantly reduced. It is hypothesized that *SlCCD1A* is the key gene for cleaving carotenoids to produce volatiles in tomato fruits.

## Figures and Tables

**Figure 1 foods-10-02678-f001:**
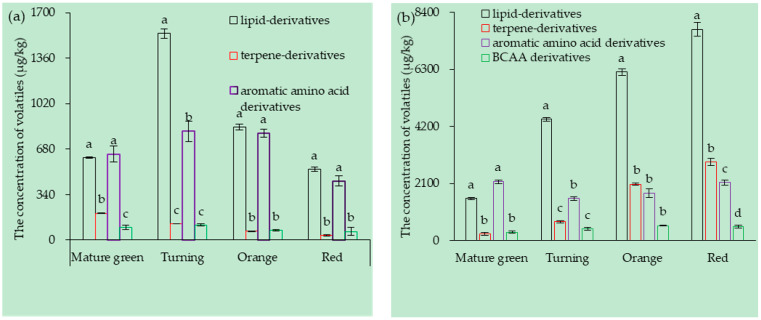
The contents of volatiles derived from different metabolic pathways in tomato fruits (μg/kg): (**a**) TI4001 and (**b**) CI1005. Lower case letters indicate significant differences in volatile contents among different metabolic pathways (*p* < 0.05).

**Figure 2 foods-10-02678-f002:**
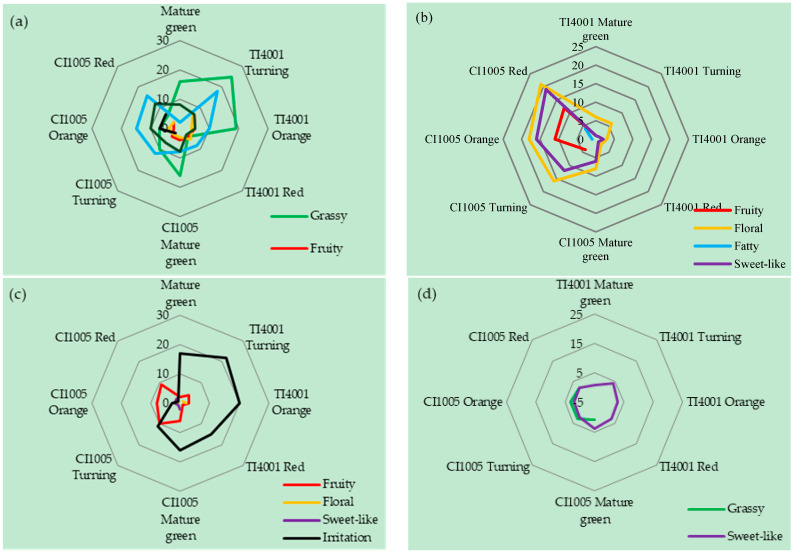
Contribution of volatiles derived from different metabolic pathways to the aromas of tomato fruits: (**a**) fatty acid-derived, (**b**) carotenoid-derived, (**c**) phenylpropane-derived, and (**d**) BCAA-derived.

**Figure 3 foods-10-02678-f003:**
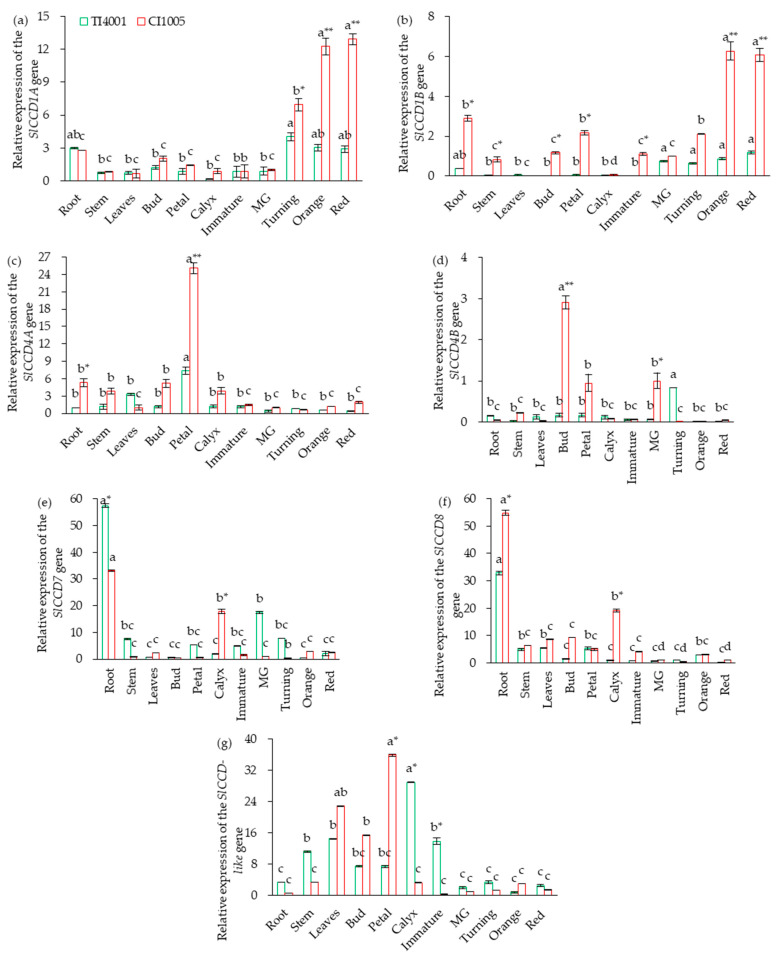
Expression of *SlCCD*s among different tissues of tomatoes. (**a**–**g**) The relative expression levels of the *SlCCD1A*, *SlCCD1B*, *SlCCD4A*, *SlCCD4B*, *SlCCD7*, *SlCCD8*, and *SlCCD-LIKE* genes. From left to right, the tissues were the root, stem, leaf, bud, petal, calyx, the immature green stage, the mature green stage, the turning stage, the orange stage, and the red stage. * Indicates a significant difference between TI4001 and CI1005 (*p* < 0.05). ** Indicates a very significant difference between TI4001 and CI1005 (*p* < 0.01). Lower case letters indicate significant differences in volatile contents among different tissues (*p* < 0.05).

**Figure 4 foods-10-02678-f004:**
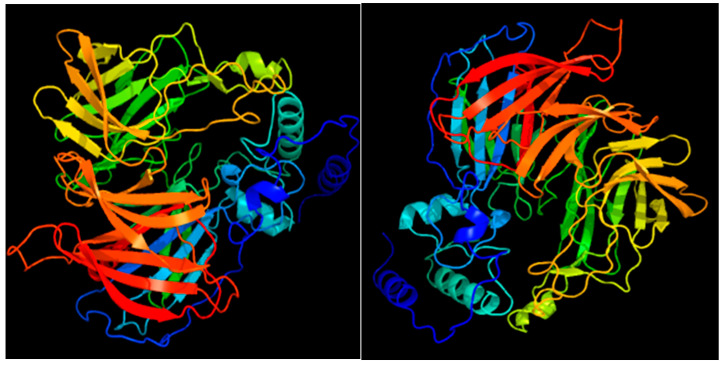
The 3D prediction structures of the SlCCD1A and SlCCD1B proteins in tomatoes.

**Figure 5 foods-10-02678-f005:**
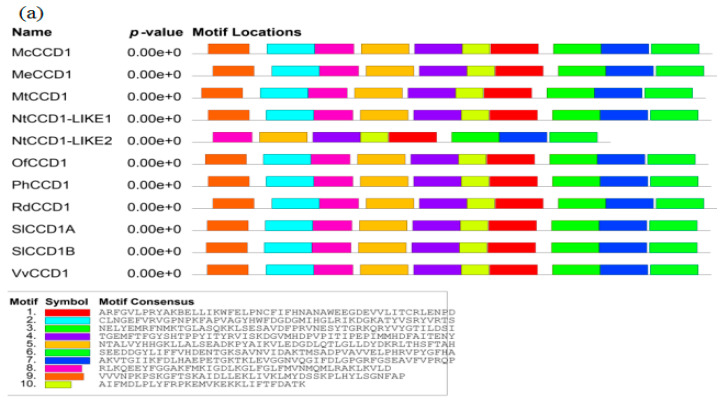
The conserved domains of the CCD1 amino acid sequences from a dicotyledonous plant: (**a**) conserved domains and (**b**) logo for the most conserved domain.

**Figure 6 foods-10-02678-f006:**
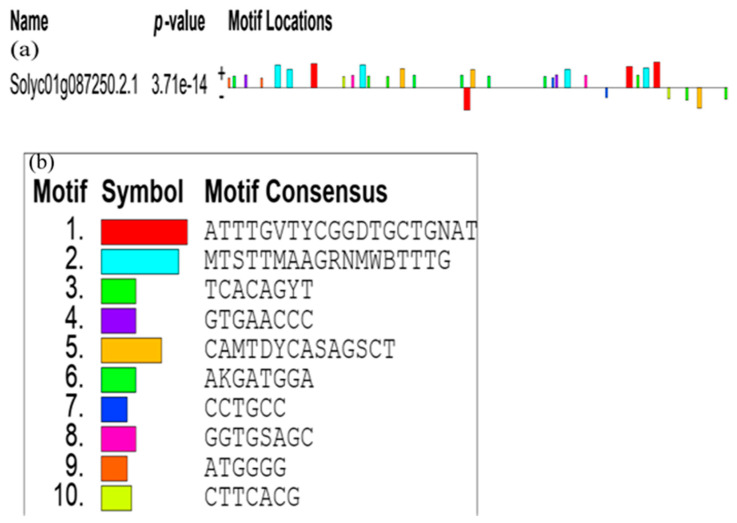
The conserved domains of the SlCCD1A amino acid sequences: (**a**) conserved domains and (**b**) amino acid sequences of the conserved domains.

**Figure 7 foods-10-02678-f007:**
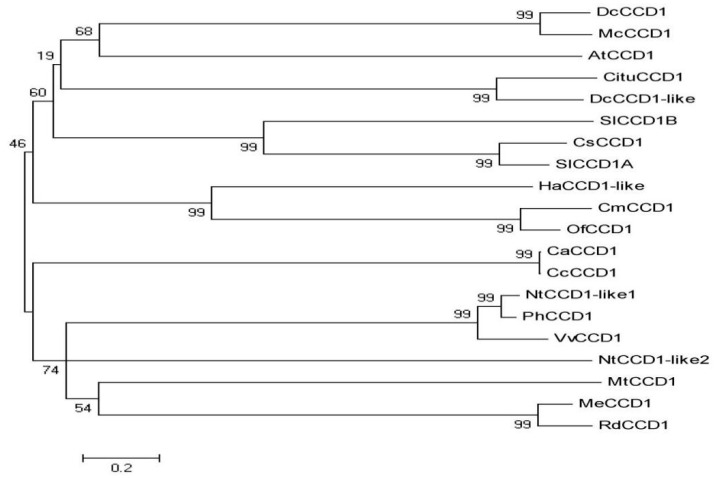
Cluster diagram of phylogenetic relationships of the CCD1s from dicotyledonous plants.

**Figure 8 foods-10-02678-f008:**
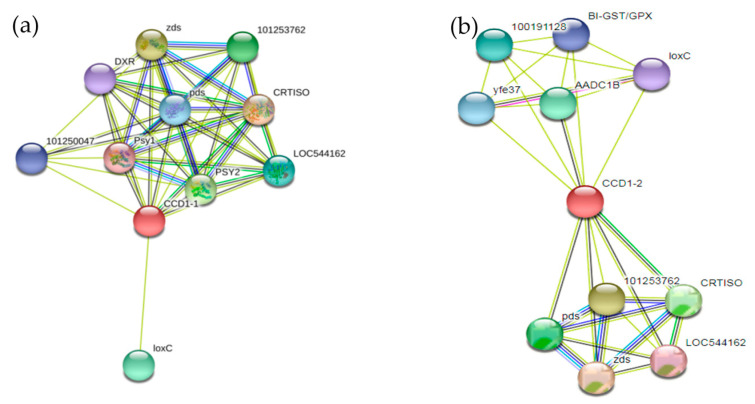
Diagram of interaction networks of the SlCCD1A and SlCCD1B proteins: (**a**) the SlCCD1A protein and (**b**) SlCCD1B protein.

**Figure 9 foods-10-02678-f009:**
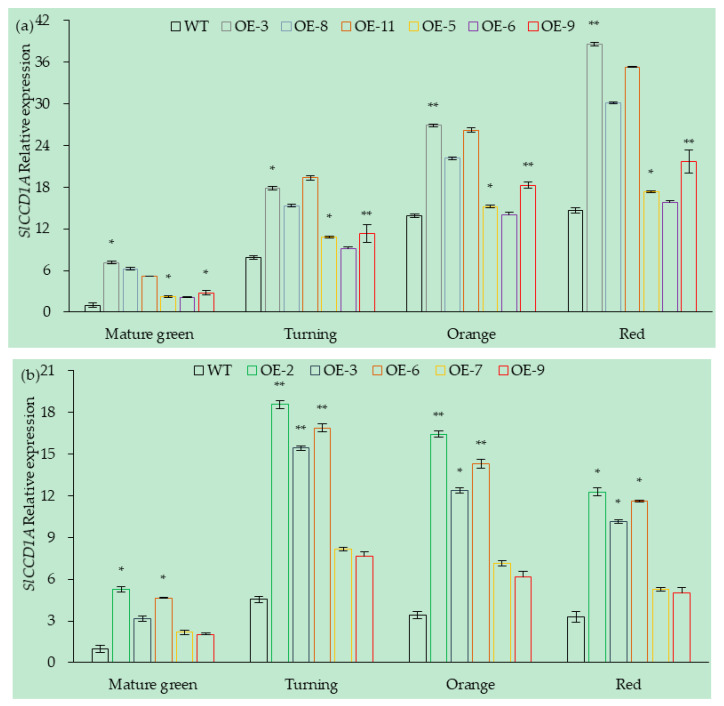
Relative expressions of *SlCCD1A* in OE tomato fruits in (**a**) CI1005 and (**b**) TI4001. WT: pVbg2307 empty vector; * indicates that *SlCCD1A* expression was significantly higher in OE lines than in WT (*p* < 0.05); ** indicates that *SlCCD1A* expression was very significantly higher in OE lines than in WT (*p* < 0.01).

**Figure 10 foods-10-02678-f010:**
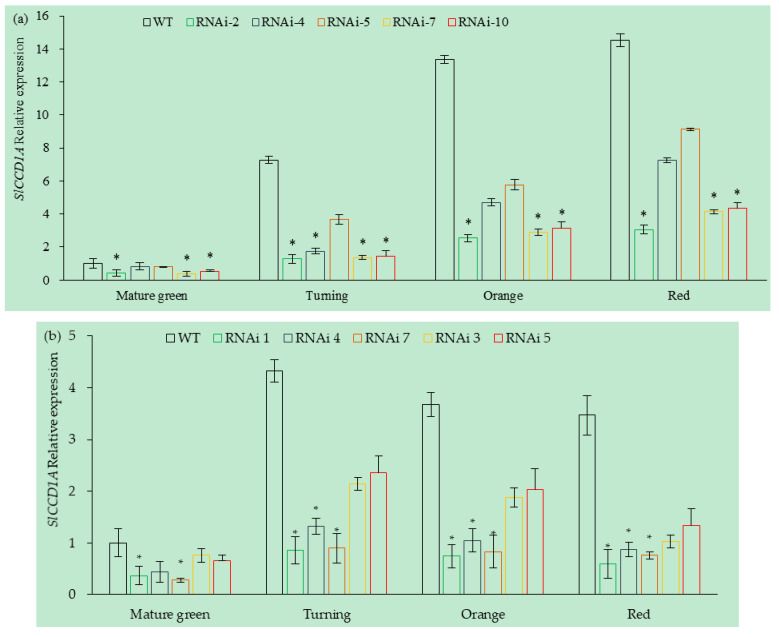
Relative expressions of *SlCCD1A* in RNAi transgenic tomato fruits in (**a**) CI1005 and (**b**) TI4001. WT: pCAMBIA1300 empty vector; * indicates that *SlCCD1A* expression was significantly lower in RNAi lines than in WT (*p* < 0.05).

**Figure 11 foods-10-02678-f011:**
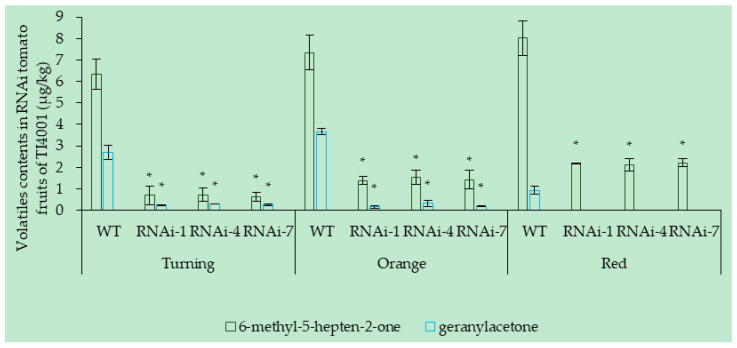
RNAi of *SlCCD1A* on the contents of the carotenoid-derived volatiles in TI4001 tomato fruits (μg/kg). * indicates that the content of 6-methyl-5-heptene-2-one or geranylacetone was significantly lower in RNAi lines than in WT (*p* < 0.05).

**Figure 12 foods-10-02678-f012:**
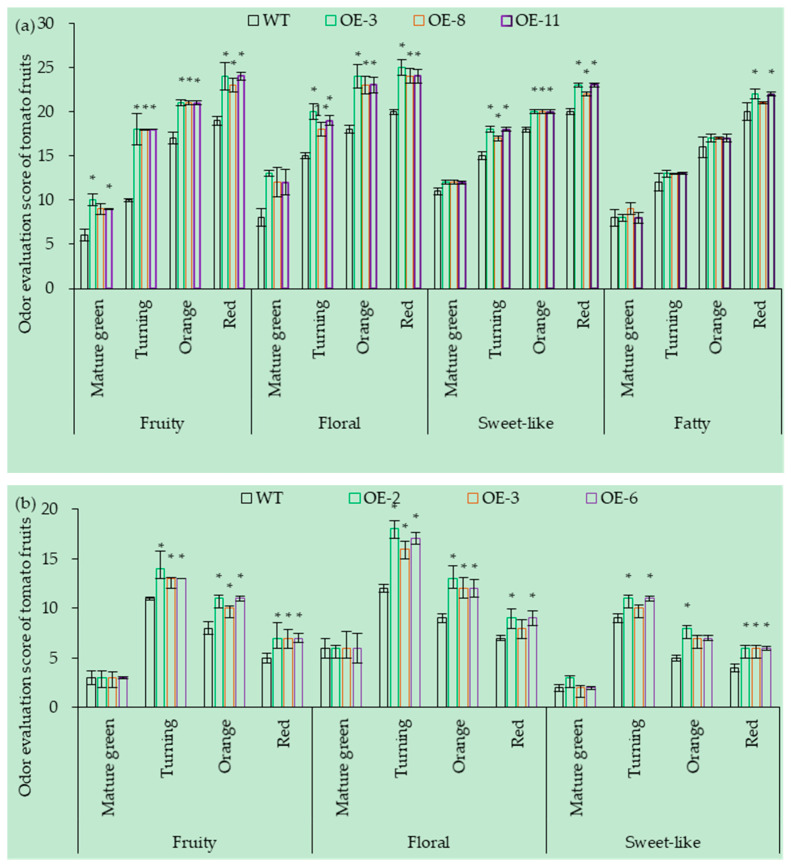
OE of *SlCCD1A* on the aroma scores of tomato fruits in (**a**) CI1005 and (**b**) TI4001. WT: pVbg2307 empty; * indicates that the aroma score was significantly higher in OE lines than in WT (*p* < 0.05).

**Figure 13 foods-10-02678-f013:**
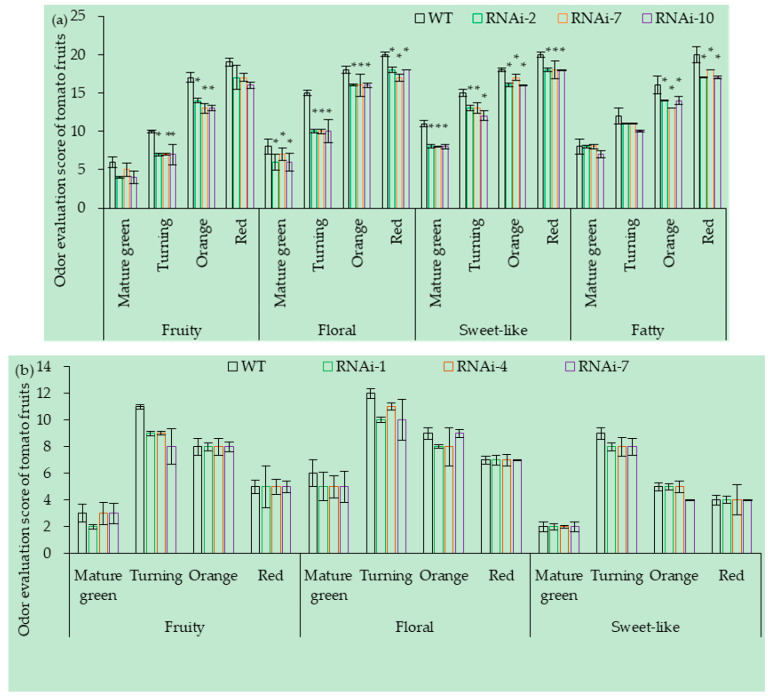
RNAi of *SlCCD1A* on the aroma scores of tomato fruits (**a**) CI1005 and (**b**) TI4001. WT: pCAMBIA1300 empty vector; * indicates that the aroma score was significantly lower in RNAi lines than in WT (*p* < 0.05).

**Figure 14 foods-10-02678-f014:**
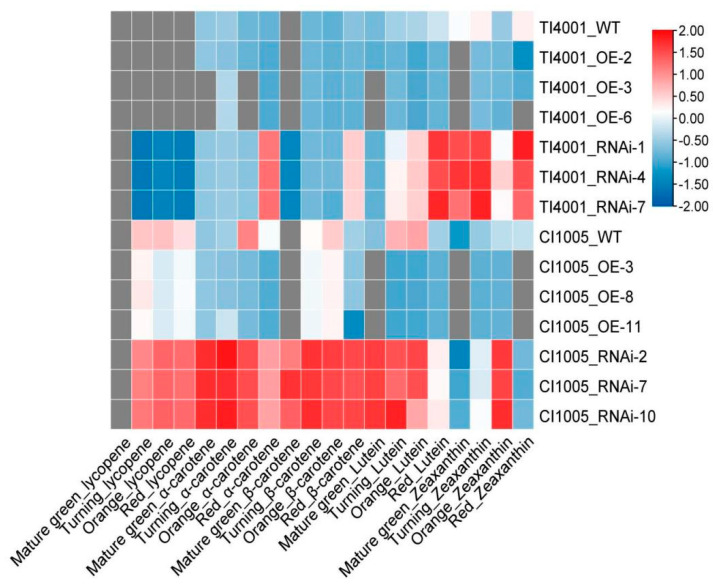
Effects of *SlCCD1A* on the carotenoid contents in tomato fruits.

**Table 1 foods-10-02678-t001:** Predictions of the transcription factors regulating the promoters of *SlCCD1A* and *SlCCD1B* in tomatoes.

	Matrix ID	Name	Family	Score	Relative Score	Strand	Predicted Sequence
*SlCCD1A and SlCCD1B*	MA0020.1	Dof2	Dof-type	8.76	1	− ^a^	aaagca
	MA0053.1	MNB1A	Dof-type	8.11	1	−	aaagc
	MA0063.1	Nkx2-5	NK-related factors	9	0.99	−	ataattg
	MA0067.1	Pax2	Paired domain only	8.36	0.97	−	tgtcatgc
	MA0095.1	YY1	More than 3 adjacent	7.39	0.95	−	tccatc
			zinc finger factors				
	MA0102.3	CEBPA	C/EBP-related factors	12.06	0.94	−	atttcatcaca
	MA0049.1	hb	Factors with multiple	10.71	0.93	−	caacaaaaaa
			dispersed zinc fingers				
	MA0046.2	HNF1A	POU domain factors	14.6	0.91	+ ^b^	aactaataatttaca
	MA0108.1	TBP	TBP-related factors	10.96	0.9	+	gtataaaattgggag
	MA0100.2	Myb	Myb/SANT domain factors	7.75	0.88	-	atagctgaca
*SlCCD1A*	MA0035.1	Gata1	GATA-type zinc fingers	6.92	0.99	+	agatgg
*SlCCD1B*	MA0036.1	GATA2	GATA-type zinc fingers	6.65	1	−	ggata
	MA0064.1	PBF	Dof-type	8.06	1	−	aaagc
	MA0024.2	E2F1	E2F-related factors	7.29	0.88	−	cgagcgggaat
	MA0050.1	IRF1	Interferon-regulatory factors	11.77	0.88	−	taaaatgaaact

Note: a. “−” indicates the antisense strand of DNA; b. “+” indicates the sense strand of DNA.

**Table 2 foods-10-02678-t002:** OE of *SlCCD1A* on the contents of carotenoid-derived volatiles in CI1005 tomato fruits (μg/kg).

Volatiles	Mature Green				Turning				Orange				Red			
	WT	OE-3	OE-8	OE-11	WT	OE-3	OE-8	OE-11	WT	OE-3	OE-8	OE-11	WT	OE-3	OE-8	OE-11
(E)-citral	—	—	—	—	6.31	22.14 *	21.35 *	21.66 *	42.32	84.44 *	70.12 *	78.55 *	33.09	105.36 *	95.3 *	106.34 *
β-cyclocitral	—	—	—	—	7.28	23.16 *	20.18 *	22.03 *	13.25	46.38 *	34.65 *	44.38 *	9.14	66.13 *	55.24 *	58.72 *
6-methyl-5-hepten-2-ol	1.77	4.11 *	3.86 *	3.94 *	1.46	16.98 *	15.32 *	15.69 *	5.68	33.17 *	28.17 *	30.57 *	6.35	44.28 *	38.17 *	40.42 *
6-methyl-5-hepten-2-one	8.71	10.69	9.25	9.87	45.86	108.14 *	101.6 *	103.12 *	139.47	357.13 *	348.19 *	348.78 *	179.15	502.32 **	478.37 **	489.32 **
geranylacetone	3.06	8.24 *	7.54 *	8.03 *	15.53	64.16 *	62.18 *	63.94 *	59.08	138.29 *	112.13 *	115.18 *	77.54	213.47 **	190.45 **	203.69 **
(E)-á-ionone	—	—	—	—	18.68	45.76 *	42.17 *	42.66 *	20.36	63.19 *	56.88 *	59.11 *	32.48	112.94 *	96.42 *	110.35 *
β-ionone	—	—	—	—	4.61	38.19 *	36.23 *	37.03 *	5.4	67.65 **	50.65 **	56.73 **	11.23	135.54 *	130.23 *	130.69 *
geraniol	—	—	—	—	12.04	35.63 *	33.19 *	33.69 *	13.16	67.18 *	65.89 *	63 *	32.07	87.27 *	80.38 *	85.33 *
3,7-dimethyl-6-octen-1-ol					—	4.36	3.18	3.24	9.28	35.36 *	30.79 *	32.11 *	17.24	52.17 *	47.88 *	53.24 *
neral	2.1	3.6	3.2	3.3	7.16	35.66 *	25.63 *	31.18 *	20.17	65.47 *	56.24 *	58.65 *	34.15	92.88 *	87.36 *	90.43 *
acrylacetaldehyde	—	—	—	—	4.28	5.69 *	5.23 *	5.44 *	18.03	37.63 *	26.69 *	28.63 *	18.01	82.14 *	76.22 *	79.38 *
(E)-farnesal	—	—	—	—	—	5.63	3.16	5.01	9.42	54.16 *	50.98 *	52.19 *	22.67	83.31 *	79.44 *	81.34 *
hexahydropseudoionone	—	—	—	—	—	14.46	12.01	13.06	—	28.17	20.18	25.77	20.15	55.42 *	50.16 *	51.07 *
pseudoionone	—	—	—	—	3.17	27.15 *	23.05 *	23.96 *	18.06	64.69 *	53.84 *	56.69 *	34.77	45.86 *	46.12 *	45.25 *
farnesyl acetone	0.47	3.69 *	3.44 *	3.58 *	5.27	48.36 *	42.36 *	45.23 *	34.17	62.65 *	54.89 *	60.45 *	48.94	75.17 *	73.99 *	76.45 *

Note: * indicates that the contents of carotenoid-derived volatiles were significantly higher in OE lines than in WT (*p* < 0.05); ** indicates that the contents of carotenoid-derived volatiles were very significantly higher in OE lines than in WT (*p* < 0.01); “—” no detected the volatile.

**Table 3 foods-10-02678-t003:** OE of *SlCCD1A* on the contents of the carotenoid-derived volatiles in TI4001 tomato fruits (μg/kg).

Volatiles	Turning				Orange				Red			
	WT	OE-2	OE-3	OE-6	WT	OE-2	OE-3	OE-6	WT	OE-2	OE-3	OE-6
6-methyl-5-hepten-2-one	6.45	28.36 *	24.18 *	27.48 *	7.27	23.44 *	18.16 *	22.16 *	7.7	30.11 *	20.43 *	32.08 *
geranylacetone	2.82	15.33 *	12.16 *	14.36 *	3.64	26.63 *	20.14 *	25.63 *	0.86	21.14 *	18.55 *	20.47 *
(E)-á-ionone	—	12.36	9.16	11.98	2.06	10.25 *	8.23 *	9.65 *	—	9.36	6.23	8.96
β-ionone	—	7.24	5.62	6.89	—	9.43	7.93	8.74	—	—	—	—
geraniol	—	2.44	1.36	2.14	—	1.12	0.71	0.97	—	0.69	0.55	0.72
neral	—	1.84	1.56	1.74	—	0.93	0.63	0.86	—	—	—	—
pseudoionone	—	2.18	1.69	2.03	—	1.33	0.86	1.23	—	—	—	—

Note: * indicates that the contents of carotenoid-derived volatiles were significantly higher in OE lines than in WT (*p* < 0.05);“—” no detected the volatile.

**Table 4 foods-10-02678-t004:** RNAi of *SlCCD1A* on the contents of the carotenoid-derived volatiles in CI1005 tomato fruits (μg/kg).

Volatiles	Mature Green				Turning				Orange				Red			
	WT	RNAi-2	RNAi-7	RNAi-10	WT	RNAi-2	RNAi-7	RNAi-10	WT	RNAi-2	RNAi-7	RNAi-10	WT	RNAi-2	RNAi-7	RNAi-10
(E)-citral	—	—	—	—	6.27	2.74	2.88	3.21	43.16	34.69	36.12	35.01	32.54	19.06 *	18.98 *	20.15 *
β-cyclocitral	—	—	—	—	7.32	2.89	3.02	3.16	12.44	3.87 *	4.02 *	4.29 *	8.16	2.78	3.22	3.18
6-methyl-5-hepten-2-ol	1.56	—	—	—	1.48	—	—	—	5.35	1.3	1.24	1.33	6.77	3.04	2.15	2.47
6-methyl-5-hepten-2-one	8.63	2.88 *	3.05 *	3.17 *	46.23	8.97 *	9.23 *	10.65 *	137.66	31.77 *	32.17 *	33.17 *	176.22	43.18 **	44.72 **	48.5 *
geranylacetone	3.28	—	—	—	15.34	2.77 *	2.87 *	3.14 *	58.44	11.26 *	11.57 *	12.33 *	76.12	16.45 *	17.35 *	18.31 *
(E)-á-ionone	—	—	—	—	18.92	4.57 *	4.64 *	4.26 *	22.17	2.55 *	2.65 *	2.88 *	30.09	5.15 *	5.7 *	6.1 *
β-ionone	—	—	—	—	4.94	1.63	1.88	2.1	5.61	—	—	—	13.05	2.99 *	3.04 *	3.17 *
geraniol	—	—	—	—	12.88	—	—	1.47 *	12.47	2.04	1.97	2.18	31.56	5.02 *	5.34 *	5.39 *
3,7-dimethyl-6-octen-1-ol	—	—	—	—	—	—	—	—	9.56	2.06	2.04	2.22c	16.88	3.14	3.22	3.16
neral	2.4	—	—	—	7.39	1.68	1.96	2.16	19.16	3.19	3.18	3.24c	33.22	7.35	7.88	7.96
acrylacetaldehyde	—	—	—	—	5.26	—	—	—	18.23	3.14	3.22	3.18c	17.66	8.04	8.35	9.2
(E)-farnesal	—	—	—	—	—	—	—	—	9.68	7.73	7.88	8.16b	21.35	2.35	2.98	3.6c
hexahydropseudoionone	—	—	—	—	0.69	—	—	—	—	—	—	—	18.44	4.5	4.65	4.26
pseudoionone	—	—	—	—	3.41	1.75	1.86	2.19	17.54	6.8	7.1	7.06	33.83	9.56 *	9.77 *	10.19 *
farnesyl acetone	0.55	—	—	—	6.18	1.98	2.04	2.36	33.18	8.33 *	8.54 *	8.63 *	45.14	5.34 *	5.57 *	5.83 *

Note: * indicates that the contents of carotenoid-derived volatiles were significantly higher in OE lines than in WT (*p* < 0.05); ** indicates that the contents of carotenoid-derived volatiles were very significantly higher in OE lines than in WT (*p* < 0.01); “—” no detected the volatile.

## Data Availability

All the date from this study are available in this article.

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
