# Peer review of "SlCCD1A Enhances the Aroma Quality of Tomato Fruits by Promoting the Synthesis of Carotenoid-Derived Volatiles"

_foods, 2021, doi:10.3390/foods10112678_

Round 1
Reviewer 1 Report
The manuscipt deals with exploration of impact of SlCCD on aroma quality of tomato fruits.
Nevertheless, the manuscript presents several shortcomings that should be modified.
1-It is well known that most important part of volatils is present in green parts (bracts, stems, branches, leaves, sepals and petioles) and may influence more sensory analyses. How the samples of fruits have been prepared? This is not present in M&M. The sepals were present on the fruits presented to the representative panel? This clearly modify the sensory.
2- the panel selected is too low and therefre not representative.
3- There are not ethical consent of the Human subjects involved in the study. This should (mandatory) added.
4-The figures should be redrawn. Figures 1, 3, 10, 11 and 12 are not readible. Please, fill the column with different colors. The colored figures are free-cost.
5- statistical analyses are not sufficient and more details should be added concerning the correlation between sensory and genes expression.
6- the conclusion are mere speculative
Further remarks
The abstract needs to be improved by adding the methods used.
Please add, in introduction, that the volatils of green part are important for sensory of fruit quality of the tomatoes.
Please use a phenological scale to define precisely the stage of tomato fruits maturation (for exampe BBCH scale).
Line 745. please explain the difference between grassy and fatty
Author Response
Drear Editor and Reviewers,
Thank you very much for your time and effort in reviewing our manuscript, “SlCCD1A enhances the aroma quality of tomato fruits by promoting the synthesis of carotenoid-derived volatiles”, manuscript ID: foods-1413725, which has been fully improved according to your opinions. Please review it again.
Response to Reviewer 1 Comments
Point 1: It is well known that most important part of volatils is present in green parts (bracts, stems, branches, leaves, sepals and petioles) and may influence more sensory analyses. How the samples of fruits have been prepared? This is not present in M&M. The sepals were present on the fruits presented to the representative panel? This clearly modify the sensory.
Response 1: Twenty tomato fruits were selected at each mature stage by uniform size, uniform coloration, no cracks, no diseases, no insects, and no decay [21]. The tomato sepals were removed. Then, tomato fruits were crushed into homogenate using a FJ200-SH homogenizer (Specimen model factory, Shanghai, China). The volatiles were measured in tomato fruits using the headspace solid-phase microextraction gas chromatography–olfactometry–mass spectrometry method. The panel only sniffed and descriped the aromas characteristics and intensity of different volatiles from olfactory detector. The tomato fruits with sepals did not present to the representative panel.
Point 2: The panel selected is too low and therefore not representative.
Response 2: Yes. Sensory evaluation does require the participation of a wide range of panel. In this study, fifty female and fifty male volunteers including native Asia, Europe, and Africa were recruited, ages from 18 to 60 years old [24]. The volunteers were trained the knowlege of food flavor chemistry and skills of tomato flavor sensory evaluation by food flavor specialist from Northwest A&F University for two weeks. Among them, five female and five male volunteers were selected by triangulation test [25] to sniff and descripe the aromas characteristics and intensity of different volatiles using olfactory detector.
The volatiles were detected and sniffed in the Horticultural Science Research Center of Northwest A&F University. Each sample was detected and sniffed for more than one hour. The test fee for each sample is 100 RMB. We conducted at least 216 tests.
Due to the limitation of test site and time, we selected 10 volunteers with outstanding odor evaluation ability from 100 trained volunteers to form the panel.The panel can sniff out more than 20 aromas from each sample.
Point 3: There are not ethical consent of the Human subjects involved in the study. This should (mandatory) added.
Response 3: Yes. The ethical consent of the Human subjects was added in 2.4— “First, the test was reviewed by the Ethics Committee of Northwest A&F University. Second, volunteers were informed in advance of the potential hazards of the sensory evaluation test. Third, volunteers only sniffed and described the volatile aromas in the tomato fruits, which is no danger to human health”.
Point 4: The figures should be redrawn. Figures 1, 3, 10, 11 and 12 are not readible. Please, fill the column with different colors. The colored figures are free-cost.
Response 4: Figures 1, 3, 10, 11 and 12 had been redrawn and filled the column with different colors.
Point 5: Statistical analyses are not sufficient and more details should be added concerning the correlation between sensory and genes expression.
Response 5: The correlation of SlCCD1A expression level was significantly positive (P < 0.05) with sensory evaluation scores of floral, fruity, sweet-like, and fatty aromes by Pearson correlation analysis.
Point 6: The conclusion are mere speculative.
Response 6: Through overexpression and RNA interference of SlCCD1A, it was found that changes in expression level of SlCCD1A could affect volatiles and carotenoids contents, as well as aroma intensity of tomato fruits. So, it is hypothesized that SlCCD1A is the key gene for cleaving carotenoids to produce volatiles in tomato fruits. Next, we will further confirm the SlCCD1A's contribution to improving tomato fruit flavor.
Further remarks
Point 7: The abstract needs to be improved by adding the methods used.
Response 7: The methods used have been added to the abstract “The volatiles contents were measured in tomato fruits using gas chromatography-mass spectrometry. The scores of tomato taste and odor characteristics were evaluated by hedonistic taste and olfaction.”
Point 8: Please add, in introduction, that the volatils of green part are important for sensory of fruit quality of the tomatoes.
Response 8: The importance of green leaf volatiles has been added to the introduction “The most important volatiles of tomato come from green parts (stems, branches, leaves, sepals and petioles). They have a green leaf aroma and are called green leaf volatiles (GLVs), such as n-hexanal, n-hexanol, (E)-2-hexenal, (E)-2-hexenol, (Z)-3-hexenal, (Z)-3-hexenol, (E)-3-hexenal, (E)-3-hexenol, (Z)-3-hexenyl acetate. GLVs can transmit signals, induce the internal defense response of plants, and enhance the adaptability of plants to adversity. GLVs can be transferred to tomato fruits, increase grassy aroma, improve fruit freshness.”
Point 9: Please use a phenological scale to define precisely the stage of tomato fruits maturation (for exampe BBCH scale).
Response 9: According to BBCH scale, the maturity stage of tomato fruits can be divided into the mature green (BBCH-Scal 79~81), turning (BBCH-Scal 82~84) , orange (BBCH-Scal 85~87), and red ripe (BBCH-Scal 88~89) stages.
Point 10: Line 745. please explain the difference between grassy and fatty.
Response 10: The grassy aroma is associated with freshness, while the fatty aroma is associated with decay.
Chain-like alcohols, aldehydes, ketones and esters increased with the number of carbon atoms in the main chain, and the aroma changed from fruity to grassy and then to fatty. The aromas of the alcohols, aldehydes, ketones, acids, and esters can change with the increase of carbon atoms numbers, C4 with caramel aroma, C6 with grassy aroma, C8 with mushroom-like aroma, C9 with cucumber-like aroma, and C10 with fatty aroma. As the carbon chain of the volatiles molecule increasing, the intensity of fatty aroma get stronger. But, the fatty aroma of C15 ~ C20 volatiles disappeared.
Thank foods for providing the review forum. Thank you very much for your time and effort in reviewing our manuscript. Your comments have been very helpful in improving our research. It's a pleasure to communicate with you.
With best regards,
Guo-ting Cheng, Yu-shun Li, Shing-ming Qi, Jin Wang, Pan Zhao, Qian-qi Lou, Ahmed H. El-Sappah,Yan-feng Wang, Xiang-qian Zhang, and Yan Liang
21th, October, 2021

Reviewer 2 Report
Dear editor and colleagues
I have read the entitled manuscript “SlCCD1A gene enhances aroma quality of tomato fruits by promoting synthesis of carotenoid-derived volatiles” with interest, and have concluded my review
It is rather a technically sound study focusing on the cross-talk mechanism regarding carotenoid derived volatiles and the SLCCD1 gene.
The authors used two differing tomato varieties (one rich vs weak aroma & volatiles), sensory nalalyses and provided data from overexpressing and knocked out genes (SlCCD1A), while following the metabolome response. According to my opinion this is a robust study.
Still, although the authors have used appropriate techniques and useful bioinformatic analyses, nevertheless the manuscript itself is lacking the clarity due to numerous grammatical, syntax and typos that makes it really difficult for the reader to comprehend. My advice to the authors is to seek professional English editing support, because the current version does not do justice to their work.
Also, the introduction is weak and does not fully explores research on the carotenoid pathways regarding volatiles.
On the contrary the materials and methods section is disproportionally extensive and I fell that there is no need to report well established protocols (for instance Agrobacterium transformation). Nonetheless since foods journal does not have a page limit it is up to the authors to decide on that.
Some minor points
Regarding the statistics, I failed to see information regarding whether the authors performed normality tests (that is a pre-requested in ANOVA)
Also, I did not see any information regarding the primers used in the study
Figure 3 legend reports significant differences (*) that are not shown in the figure
Figure 4 does not offer significant information. remove it or use it as a supplementary file
Overall, even though I can appreciate that this is an interesting and extensive work, I still have the feeling that it is roughly presented and did not receive proofreading (for instance there are numerous times that the term RANi is reported throughout the manuscript).
Based on the above reasons my recommendation is a major restructure and revision
Author Response
Drear Editor and Reviewers,
Thank you very much for your time and effort in reviewing our manuscript, “SlCCD1A enhances the aroma quality of tomato fruits by promoting the synthesis of carotenoid-derived volatiles”, manuscript ID: foods-1413725, which has been fully improved according to your opinions. Please review it again.
Response to Reviewer 2 Comments
Point 1: The manuscript itself is lacking the clarity due to numerous grammatical, syntax and typos that makes it really difficult for the reader to comprehend. My advice to the authors is to seek professional English editing support, because the current version does not do justice to their work.
Response 1: We carefully corrected the grammatical, syntax and typos in this manuscript. Then, we seeked to the MDPI Author Services for English Editing Services (ID: English-35804. https://www.mdpi.com/authors/english.). Besides, Dr. Tayeb Muhammad from Pakistan and Dr. Ahmed.H.El-Sappah from Egypt revised the English language in this manuscript.
Point 2: The introduction is weak and does not fully explores research on the carotenoid pathways regarding volatiles.
Response 2: The contents of green leaf volatiles and carotenoid-derived volatiles were added in the introduction, and related references were added. SlCCDs enzymes can cleave carotenoids at different double bond positions and produce a variety of volatiles.
Point 3: On the contrary the materials and methods section is disproportionally extensive and I fell that there is no need to report well established protocols (for instance Agrobacterium transformation). Nonetheless since foods journal does not have a page limit it is up to the authors to decide on that.
Response 3: Thank you. Some well established protocols were removed from the materials and methods section and related references were added.
Point 4: Regarding the statistics, I failed to see information regarding whether the authors performed normality tests (that is a pre-requested in ANOVA).
Response 4: The significant difference among samples was performed by a one-way ANOVA (P < 0.05) after the homogeneity of variance test.
Point 5: I did not see any information regarding the primers used in the study.
Response 5: The primers used in the study were listed in table S1.
Point 6: Figure 3 legend reports significant differences (*) that are not shown in the figure.
Response 6: The SlCCDs expressions with significant differences were added (*) between accessions TI4001 and CI1005 in Figure 3. The significant differences of SlCCDs expressions among different tissues of the same accession were represented by a, b and c.
Point 7: Figure 4 does not offer significant information. remove it or use it as a supplementary file
Response 7: Yes. Figure 4 was removed.
Point 8: Overall, eventhough I can appreciate that this is an interesting and extensive work, I still have the feeling that it is roughly presented and did not receive proofreading (for instance there are numerous times that the term RANi is reported throughout the manuscript).
Response 8: Thank you! We proofread the manuscript carefully again. All the misspellings (for example RANi) have been corrected (RNAi).
Thank foods for providing the review forum. Thank you very much for your time and effort in reviewing our manuscript. Your comments have been very helpful in improving our research. It's a pleasure to communicate with you.
With best regards,
Guo-ting Cheng, Yu-shun Li, Shing-ming Qi, Jin Wang, Pan Zhao, Qian-qi Lou, Ahmed H. El-Sappah,Yan-feng Wang, Xiang-qian Zhang, and Yan Liang
21th, October, 2021

Round 2
Reviewer 1 Report
Dear Authors,
Thank you for considering positively the remarks made in my previous review.
Reviewer 2 Report
The authors have provided the appropriate changes and the revised manuscript has been greatly improved
Therefore, i recommend that the manuscript should be accepted
This manuscript is a resubmission of an earlier submission. The following is a list of the peer review reports and author responses from that submission.